# Ion sieving in graphene oxide membrane enables efficient actinides/lanthanides separation

Zhipeng Wang[1,2], Liqin Huang[1,2], Xue Dong[1], Tong Wu[1], Qi Qing[1], Jing Chen[1], Yuexiang Lu [1]✉ & Chao Xu [1]✉

Separation of actinides from lanthanides is of great importance for the safe management of nuclear waste and sustainable development of nuclear energy, but it represents a huge challenge due to the chemical complexity of these f-elements. Herein, we report an efficient separation strategy based on ion sieving in graphene oxide membrane. In the presence of a strong oxidizing reagent, the actinides (U, Np, Pu, Am) in a nitric acid solution exist in the high valent and linear dioxo form of actinyl ions while the lanthanides (Ce, Nd, Eu, Gd, etc.) remain as trivalent/tetravalent spheric ions. A task-specific graphene oxide membrane with an interlayer nanochannel spacing between the sizes of hydrated actinyl ions and lanthanides ions is tailored and used as an ionic cut-off filter, which blocks the larger and linear actinyl ions but allows the smaller and spheric lanthanides ions to penetrate through, affording lanthanides/actinides separation factors up to ~400. This work realizes the group separation of actinides from lanthanides under highly acidic conditions by a simple ion sieving strategy and highlights the great potential of utilizing graphene oxide membrane for nuclear waste treatment.

The sustainable development of nuclear energy relies greatly on the safe management of nuclear waste. The spent fuel unloaded from nuclear reactors consists of a series of actinides (Ans) such as U, Pu, Np, Am, and Cm, as well as a variety of fission products. To maximize the recovery of usable nuclear resources and minimize the long-term hazard of nuclear waste, an advanced nuclear fuel cycle based on the so-called "partitioning and transmutation (P/T)" strategy has been proposed around the world[1,2]. In the P/T strategy, all the long-lived actinides (U, Pu, Np, and Am) are required to be partitioned and then recycled in nuclear reactors or undergo transmutation in fast reactors or accelerator-driven systems (ADS). At the core of this strategy is the separation of these long-lived actinides from fission products, particularly the lanthanides (Lns)[3,4], because these 4f-elements are the major components of fission products and possess high neutron adsorption cross-sections that would significantly reduce the transmutation efficiency of the actinides.

However, the separation of actinides from lanthanides is quite challenging, because both the actinides and lanthanides are f-block elements and exhibit great chemical complexity. In addition, extreme conditions such as high acidity (e.g., 3 M nitric acid) and strong ionizing radiation field (α, β, γ, and neutron irradiations, etc.) further exacerbate the challenge. Although the recovery of U and Pu has been well realized in the renowned PUREX process[5], the effort to recover Np and Am has encountered great difficulties. In this regard, a so-called Group ActiNide EXtraction (GANEX) concept has been proposed to recover U through Am simultaneously from highly acidic spent fuel solution in a single-step separation, aiming towards the homogeneous recycling of these actinides in a future closed fuel cycle[6]. A number of techniques relying on solvent extraction have been pursued to realize this separation concept[7–9]. To date, the most explored one is the EURO-GANEX process, which is based on the co-extraction of actinides in a variety of oxidation states (U(VI), Pu(IV), Am(III), etc.) along with

[1]Institute of Nuclear and New Energy Technology, Tsinghua University, 100084 Beijing, China. [2]These authors contributed equally: Zhipeng Wang, Liqin Huang. ✉e-mail: luyuexiang@mail.tsinghua.edu.cn; xuchao@tsinghua.edu.cn

trivalent lanthanides from the aqueous feed solution by diglycolamide ligands and followed by selective stripping of actinides from the lanthanides with soft N-donor ligands[10,11]. Despite significant progress has been made in the development of the EURO-GANEX process, demerits such as the involvement of a large number of extraction/scrubbing/ stripping stages in tandem and the necessity to meticulously control the chemical conditions of both the organic and aqueous solutions may hamper its industrial application. An alternative approach to realize the group separation is to oxidize the actinides together into their high valent oxidation states and then separate them from the lanthanides[7,12,13]. Apart from the intrinsically stable U(VI), transnuranic elements Np, Pu, and Am can also be oxidized to the +V or +VI oxidation states and exist as linear actinyl ions ($[O=An^V=O]^+$ or $[O=An^{VI}=O]^{2+}$) under strong oxidizing conditions. In contrast, the lanthanides remain mainly in their +III oxidation states except for Ce that could exist stably as Ce(IV). The significant difference between the high valent actinides and the trivalent/tetravalent lanthanides in terms of both charge density and steric configuration offers an opportunity for efficient Ans/Lns separation in principle[7,14–17]. Unfortunately, the unstable nature of high valent Am poses a great challenge for this separation approach in practical applications because both Am(V) and Am(VI) can be readily reduced to Am(III) (the reduction potentials are +1.7 V and +1.74 V vs NHE for Am(VI)/ Am(III) and Am(V)/Am(III), respectively) by organic reagents used for separation or even by self-radiolysis product, thus making the separation of high valent Am along with other actinides from lanthanides impractical in previous studies[14–16,18]. Therefore, actinides/ lanthanides group separation still represents a seminal challenge.

It is well-known that porous carbon materials with a narrow distribution of pore/tunnel sizes, particularly in the sub-nanometric range, are of great potential for utilization in separation technologies[19–21]. Among these materials, multiple-layered graphene oxide membranes (GOMs) made of graphene oxide nanosheets have received significant attention in recent years because they are mechanically robust, size-controllable, and easy in fabrication[22–24]. GOMs with unique nanolayer structures have been synthesized and their interlayer spacing can be tuned through approaches, such as physical confinement and cation intercalation[25,26]. The tunable size effect endows these GOMs with advantages in applications such as nanofiltration and desalination.

Herein, we present a novel strategy to address the challenge of actinides/lanthanides group separation under highly acidic conditions through ion sieving in GOMs. In this strategy (Fig. 1a), a solution containing various actinides (U, Np, Pu, and Am) and lanthanides is treated with highly oxidizing reagents, by which all the actinides are oxidized to the linear dioxo actinyl ions while the lanthanides (Ce, Nd, Eu, Gd etc.) remain as spheric ions. In this case, the two groups of elements exhibit great differences in steric configuration and a distinctive gap in ionic size (Fig. 1b), and mutual separation of the actinyl ions and the lanthanide ions is then achieved by sieving through a GOM with specific pore/tunnel sizes sitting between the ionic size gap.

## Results and discussion

### Synthesis and characterization of GOM

Recently, we have demonstrated that highly acidic condition is beneficial for restraining the swelling of GOM, and its interlayer spacing can be accurately adjusted in a wide range (11.4–15.5 Å) by controlling the oxidation degree of the GO nanosheets as well as the acidity of the solution[27]. In the present work, a GOM with an interlayer spacing ($d$) of 13.9 Å in 3.0 mol/L HNO₃ solution was prepared as the ionic sieving filter (Detailed synthesis protocols are provided in the Methods section). The GOM has been systematically characterized with a number of techniques (SEM, AFM, XPS, XRD, Raman, etc.), and the results are shown in Fig. 2 and Supplementary Figs. 1–4. Generally, GO nanosheets (Fig. 2a) in size of micrometer scale were deposited to form the GOM

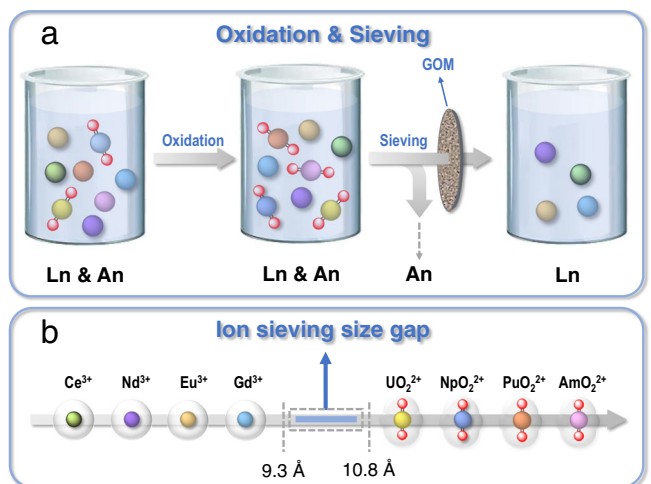

**Fig. 1 | Actinides/lanthanides group separation through ion sieving in the membrane. a** Schematic diagram for actinides/lanthanides group separation based on the ion-sieving strategy. **b** Representative lanthanide and actinide ions (details for the size calculation and relevant discussion are provided in Supplementary Note 1).

(Fig. 2b–d) on a porous nuclear track membrane, which provides mechanical support for GOM and meanwhile has abundant large pores/tunnels to allow the quick pass of solutions. The average thickness of the GOM is $150 \pm 10$ nm (Fig. 2c). X-ray photoelectron spectroscopy (XPS) analysis (Fig. 2e) of the GOM gives a C/O ratio of 2.45, which indicates the oxidation degree of the GO is relatively low and the corresponding nanosheet may consist of large areas of un-oxidized regions on its surface[28]. The dry GOM with a narrow interlayer spacing (8.4 Å) will swell up to become the wet GOM with the desired interlayer spacing (13.9 Å) in 3.0 mol/L HNO₃ (Fig. 2f). The effective nanochannel size ($\mu$) of this wet GOM can be calculated as 10.5 Å ($\mu = d - \epsilon$, where $\epsilon \approx 3.4$ Å is the thickness of the un-oxidized region of the monolayer GO nanosheet[23]), which sits in the size gap between the hydrated actinyl ions and lanthanide ions (Fig. 1b) and thus is expected to exhibit an ion-sieving effect for the two groups of ions. It should be noted that there are still debates on the exact ion-sieving mechanism in GOM. For example, the effective nanochannel size of GOM can be calculated in different ways and the ionic size of actinides/lanthanides depends greatly on their hydration or complexation status. We will present more discussions on these issues along with the sieving mechanism elucidation later in this article.

### Ion sieving of lanthanides and actinides in GOM

We adopt a permeation manner for the separation experimentally. In brief, the GOM is fixed vertically between two solution compartments (Fig. 3a), one is the feed compartment (FC) filled with a solution of actinides/lanthanides in 3.0 mol/L HNO₃ and the other is the receiving compartment (RC) filled with a solution of 3.0 mol/L HNO₃. The ion-sieving performance of GOM was first tested without the addition of oxidizing agents. As shown in Fig. 3b, c, all the lanthanide ions in the FC could pass through the GOM into the RC appreciably. For the actinides, U and Np are mostly retained in the FC, while Pu and Am could also pass through the GOM into the RC along with the lanthanide ions. Such an observation is in good accordance with the well-known redox properties of these actinides, i.e., U, Np, Pu, and Am exist mainly in the form of $UO_2^{2+}$, $NpO_2^+/NpO_2^{2+}$, $Pu^{4+}$, and $Am^{3+}$, respectively, under common acidic conditions in an aqueous solution[29,30]. In this case, the larger and linear $UO_2^{2+}$ and $NpO_2^+/NpO_2^{2+}$ ions are blocked by the GOM, but the smaller and spheric $Pu^{4+}$ and $Am^{3+}$ ions penetrate through the GOM along with the lanthanide ions, leading to an overall poor An/Ln group separation.

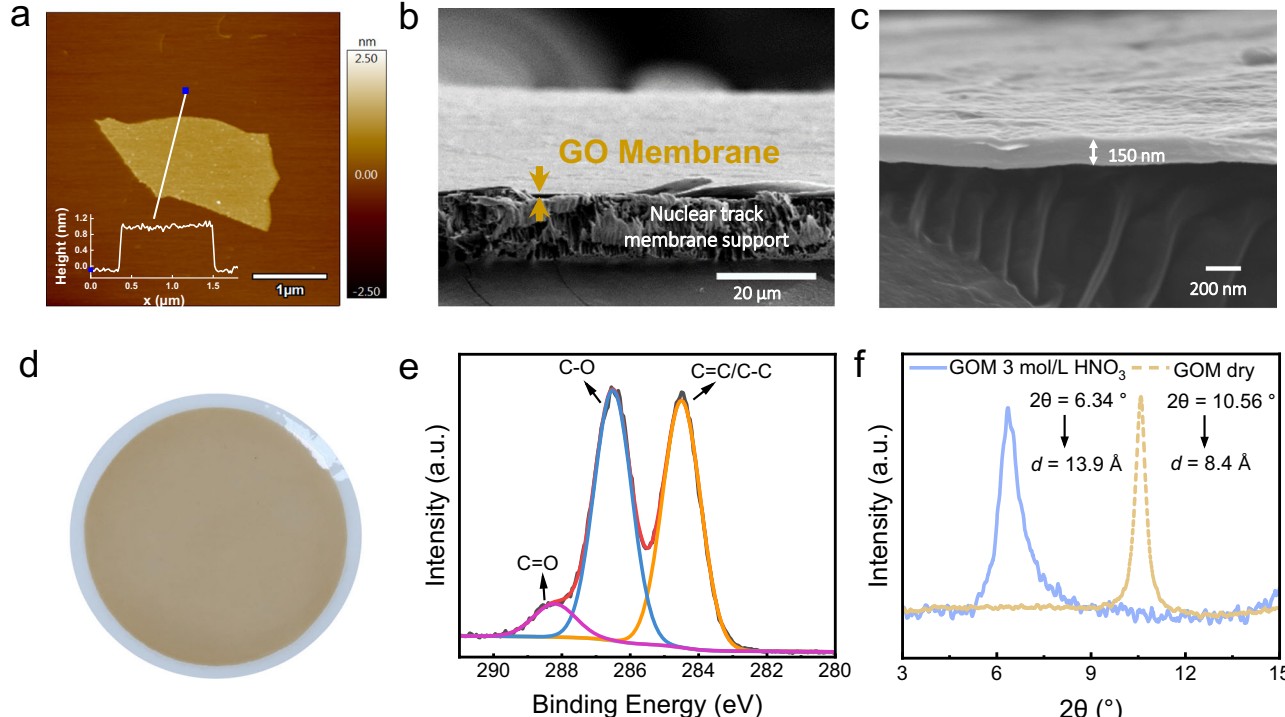

**Fig. 2 | Characterization of GO nanosheet and GOM. a** AFM image of GO nanosheet, inset is the height profile across the white line. **b** Cross-sectional SEM image of the GOM on the support membrane. **c** Cross-sectional SEM image of the GOM. **d** Digital photo of the as-synthesized GOM. **e** XPS spectra of GOM. **f** XRD patterns of the dry GOM and the wet GOM after immersing in 3.0 mol/L $HNO_3$.

For the above ion sieving, it should be noted that the highest permeation percentage (*Pct.*, defined as the percentage of concentrations of a specific element permeated from FC to RC) for the lanthanides could only reach 50% theoretically (with equal volumes of solution on both sides of the GOM) under natural diffusion conditions and the permeation kinetics is also quite slow (Fig. 3b). Therefore, to aid with the permeation of lanthanides into the RC, an organic solvent (*n*-dodecane) containing a functional ligand TODGA (*N,N,N′,N′*-tetraoctyl diglycolamide) is added into the RC to construct a confined solvent extraction system (Fig. 3d). The TODGA ligand is well-known for its excellent ability to extract trivalent/tetravalent lanthanides or actinides in nuclear waste treatment[31,32] and it has also been used as a carrier extractant in supported liquid membrane or polymer inclusion membrane for the recovery of lanthanides and actinides[33–35]. In the presence of the TODGA extraction system in the RC, trivalent and tetravalent lanthanides will be extracted into the organic phase and the concentration of lanthanides is expected to decrease significantly in the aqueous phase in the RC, thus increasing the concentration gradient across the membrane and providing an extra chemical driving force for the permeation of lanthanides through the GOM. As shown in Fig. 3e, f, significantly higher permeation percentages and faster permeation kinetics for lanthanides have been achieved in the presence of the extraction system. The permeation percentage of the lanthanide ions can reach as high as 95% in 24 h. At the same time, the permeation percentage of U and Np does not show obvious change and stays at low levels (<10%), indicating the effectiveness of sieving in the GOM.

Obviously, although the results in Fig. 3e, f prove the efficient ion sieving of U and Np from the lanthanides by the GOM, the other two actinides Pu and Am largely penetrate together with the lanthanides into the RC, emphasizing the necessity of oxidation in the FC to separate Pu and Am from the lanthanides. To address this issue, sodium bismuthate ($NaBiO_3$), known as a highly oxidizing agent capable of oxidizing Am(III) to Am(VI) quantitatively in nitric acid solutions, was employed as the chemical oxidant in this work to oxidize the actinides[12,13,17,36]. Absorption spectra (Supplementary Fig. 5) before and after the oxidation confirm the successful oxidation of Np(V), Pu(IV) and Am(III) into the hexavalent actinyl ions ($NpO_2^{2+}$, $PuO_2^{2+}$, $AmO_2^{2+}$) with $NaBiO_3$. In the presence of $NaBiO_3$ in the FC (Fig. 3g), all the actinides (U, Np, Pu, and Am) are mostly retained in the FC and the highest permeation percentage of these actinides is less than 10% over a period of 24 h (Fig. 3h, i). On the other hand, the presence of the $NaBiO_3$ oxidant in the FC shows a negligible influence on the permeation of lanthanide ions (Supplementary Fig. 6), confirming that the lanthanides remain as smaller spheric ions in the solution. These results suggest that the actinides can be well separated from the lanthanides through ion sieving in the GOM. It is worth noting that the low permeation percentage of all four actinides over a long period is encouraging since previous attempts for group An/Ln separation based on oxidation largely failed due to the fast reduction of Am(VI) by the organic reagents used for separation[14–16,37,38]. In the present work, although organic reagents are used in the RC, the oxidized Am ions in the FC have no chance to contact with these reagents owing to the blocking effect of GOM, thus avoiding the reduction issue (Supplementary Fig. 7 and Supplementary Table 1). Moreover, the constant existence of $NaBiO_3$ in the FC also helps keep the Am ions at high oxidation states during the permeation process (Supplementary Fig. 7).

## Separation demonstration with mixed Ans/Lns waste

To further assess the applicability of our sieving technique for genuine waste treatment, we explored the separation of a mixed actinides/lanthanides waste solution (Supplementary Table 2) by sieving through the GOM using the setup shown in Fig. 3g. As shown in Fig. 4, the permeation percentage of all the lanthanides reaches >90%, while the permeation percentage of the actinides remains at low levels (<10%). Herein we define the separation factor of lanthanide to actinide ($SF_{Ln/An}$) as $D_{Ln}/D_{An}$ ($D$ is the distribution ratio of a given element in the RC to the FC and it can be calculated by $Pct./(1-Pct.)$). The $SF_{Ln/An}$ values

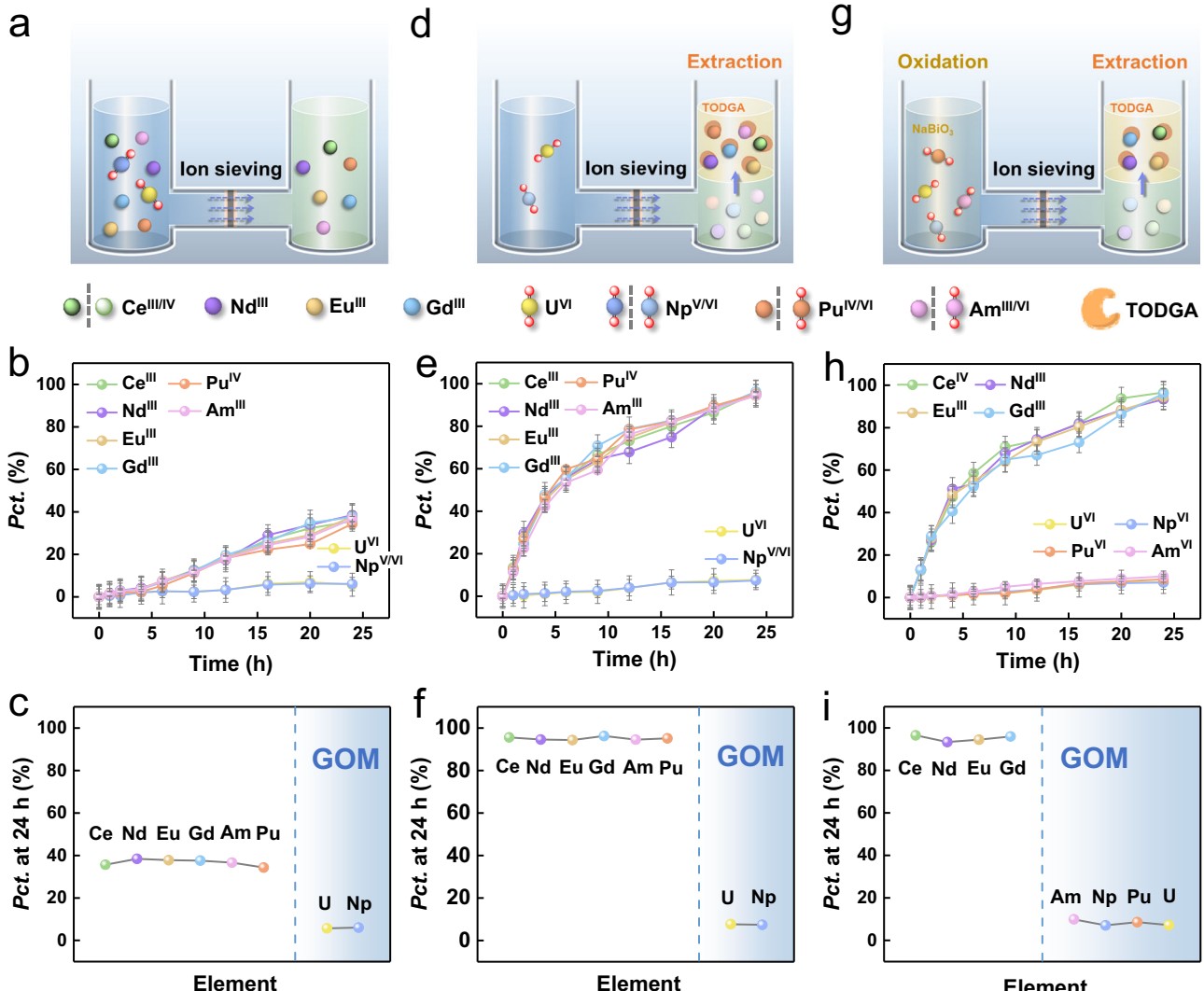

**Fig. 3 | Sieving of lanthanide and actinide ions in GOM. a–c** Setup for sieving by natural permeation and the permeation percentages (*Pct.*) of lanthanides and actinides as a function of time and elements. **d–f** Setup for sieving through the assistance of extraction and the permeation percentages (*Pct.*) of lanthanides and actinides as a function of time and elements. **g–i** Setup for sieving through the assistance of oxidation/extraction and the permeation percentages (*Pct.*) of lanthanides and actinides as a function of time and elements. These sieving experiments are conducted for each ion individually. Note: the valence sign for each ion only indicates the main oxidation state under the desired condition, other oxidation states may also exist during the ion-sieving process. All the error bars represent the standard deviation of the experiments.

for the mutual separation of all the actinides from the lanthanides all exceed 100, with the highest $SF_{Ln/An}$ value reaches ~400 for U/Ce (Fig. 4). The foregoing results clearly highlight the effectiveness of the ion-sieving strategy for group actinides/lanthanides separation in this work. Notably, the overall An/Ln separation factors in this work are mostly higher than the single-stage separation factors of An/Ln group separation using the extraction method[9,39,40] and are much higher than those using the crystallization or precipitation method[12,13,41].

Furthermore, in the presence of both TODGA extraction and NaBiO$_3$ oxidation, the ion sieving results in GOM display permeation trends of Ce > Gd > Eu > Nd for the lanthanides and Am > Pu > Np ~U for the actinides (Figs. 3i and 4). The former trend is closely related to the extraction affinity to lanthanides by TODGA. It is well-known that TODGA shows higher affinity to heavier lanthanides than lighter ones and tetravalent lanthanides than trivalent ones[31]. Consequently, among the four lanthanide ions studied in our work, Ce(IV) is the most extractable ion, followed by Gd(III), Eu(III), and Nd(III) in sequence. And such a trend is well reflected in the final order of permeation percentage of these

lanthanides. On the other hand, while the permeation of all the actinides is mostly caused by the defects and nonideal stacking of GO nanosheets in the GOM as we will discuss later in this article, the relative permeation order for the four actinides is likely related to their intrinsic redox properties. Am(VI) is the least stable one among the four actinyl ions, and even under strong oxidizing conditions it might be slightly reduced to Am(III), which can pass through the GOM and be extracted by TODGA. Pu(VI) is also not very stable and its reducing product Pu(IV) can pass through the GOM along with the lanthanides. Np(VI) is even less stable than Pu(VI)[42], but its reducing product Np(V) is still an actinyl ion that can be well blocked by the GOM (Fig. 3c, f). Therefore, the permeation of Am and Pu into the receiving solution is relatively high while the permeation of Np is well comparable to that of U. Nevertheless, the permeation percentages of the four actinides only differ slightly, implying that all the four actinides are well maintained in their actinyl form in the FC during the permeation process with the help of NaBiO$_3$ oxidation.

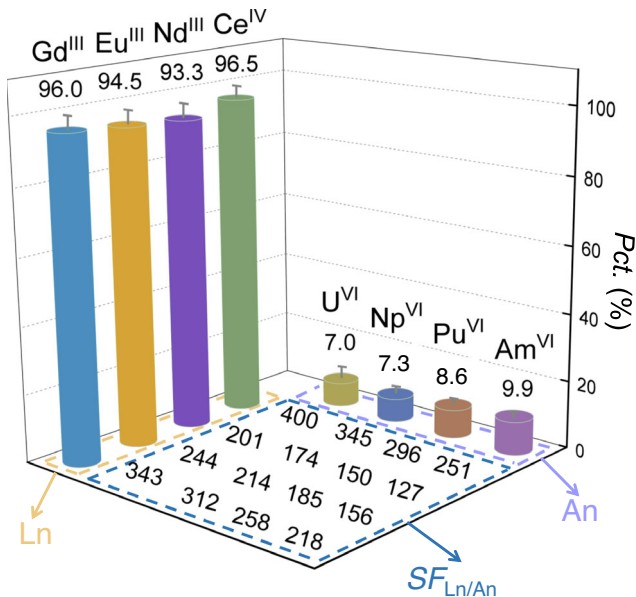

**Fig. 4 | Separation results of a mixed actinides/lanthanides waste solution after sieving through the GOM.** The solution consists U, Np, Pu, Am, Ce, Nd, Eu, and Gd in 3.0 M HNO₃ (detailed compositions of the solution are provided in Supplementary Table 2); sieving time: 24 h. All the error bars represent the standard deviation of the experiments.

## Robustness of GOM

The structural stability of GOM against ion permeation was first investigated. The structure features of GOM remain unchanged after 24 h of permeation test for Ln and An as evidenced by XRD analysis (Supplementary Fig. 8). The influence of the strong oxidizing NaBiO₃ on the performance of GOM was also evaluated. The solid NaBiO₃ in the FC would not deposit on the GOM and clog the nanochannels inside the GOM (Supplementary Figs. 9 and 10), and it has no impact on the permeation of redox-inert ions such as Nd(III), Eu(III), Gd(III), and U(VI) (Supplementary Fig. 11). More importantly, the GOM in this work exhibits outstanding resistance toward high-energy ionizing radiation under highly acidic conditions. The γ-irradition up to the dose of 100 kGy on the GOM in 3.0 mol/L HNO₃ imposes no apparent influence on the GOM structure (Supplementary Fig. 12) as well as its performance in ion sieving (Supplementary Fig. 13). Such a superior radiation-resistance under highly acidic conditions provides a substantial prerequisite for the application of GOM in real nuclear waste treatment.

## Mechanistic insights

As mentioned earlier, debates remain on the exact ion-sieving mechanism in GOM. One key debate is about the microscopic structure of nanochannels in the GOM. Many previous studies for ion sieving in GOM are mainly based on the assumption that a network of nanocapillaries in hydrated state forms between the pristine (un-oxidized graphite) regions of GO sheets and these nanocapillaries can only accept ionic species of a certain size that could fit in[23,26]. This assumption requires the existence of large areas of pristine regions in GO sheets to make sure the nanocapillaries can be connected within the GOM. However, the area of pristine regions in GO sheets varies apparently for different GOs. The local chemical structure of a typical GO has once been determined by ultra-high-resolution TEM and the results suggest there are only small and disconnected prinstine regions in that GO[43]. In this case, the chance for connected nanocapillaries between prinstine regions is quite slim. On the other hand, relatively larger and connected prinstine regions have also been observed on other GO sheets and these regions can even be further enlarged by

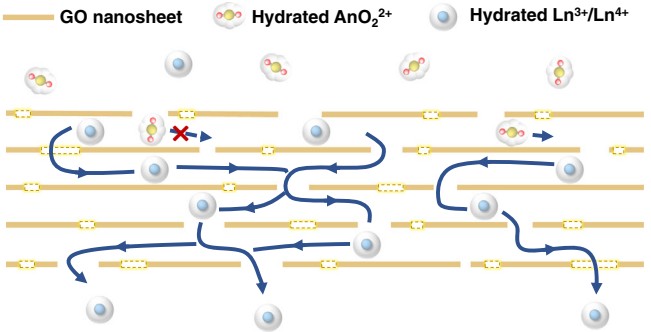

**Fig. 5 | Sectional view of possible permeation routes for lanthanides/actinides by ion sieving in GOM.** The dotted squares represent the pinholes on the GO sheets. The arrows show the possible permeation routes in the GOM.

thermal anealing[28,44]. In this work, we have deliberately controlled the synthesis conditions to lower the oxidation degree of GO[27]. Therefore, the presence of connected nanocapillaries between prinstine regions in the GOM is possible and these nanocapillaries with a size of ~10.5 Å could serve as nanochannels for the transport of lanthanide ions. Nevertheless, the real function of the oxidized regions and un-oxidized graphite regions in water/metal permeation might be very complex, so unraveling the exact transport pathway of metal ions in the GOM requires further experimental and theoretical efforts in future studies.

Another important issue is about the chemical speciation of lanthanide/actinide ions when they transport through the GOM. It is well-known that GO can adsorb both trivalent lanthanides and hexavalent actinyls through direct bonding between the functional O-bearing groups and the metals[45–47], but the adsorption effect of GO in this work is negligible under highly acidic conditions (Supplementary Fig. 14). Therefore, it is unlikely that the ions permeate through the GOM via the assistance of GO complexation. Owing to the possible availability of pristine nanocapillaries (~10.5 Å) in the GOM as we suggested above and the appropriate size gap between lanthanides ions and actinyl ions in the hydrated form (details for the size calculation and relevant discussion are provided in Supplementary Note 1), we consider the hydrated ions as the most plausible species passing through the GOM. However, it should be noted that the exact size effect alone cannot fully explain the nonideal rejection of the actinyl ions, which can still pass through the GOM but at a much lower rate as compared to lanthanides. We attribute this nonideal rejection mainly to the intrinsic structural feature of the GOM. First, it is well-known that pinhole defects exist on GO sheets and stacking voids are present in GOM due to the imperfect stacking of GO sheets[43,48], and these defects and voids are usually large in size and thus will enhance the liquid transport through the GOM and provide additional pathways for the permeation of larger ions such as actinyl in the GOM. Second, the non-uniform distribution of interlayer spacing in the GOM as evidenced by the broadened XRD peak (Fig. 2f) would also contribute to the permeation of actinyl ions. The existence of interlayer channels larger in size than the averaged values calculated from XRD is highly plausible and these channels could serve as permeation pathways for the relatively larger actinyl ions. Moreover, the linear configuration of actinyl ions might be another factor to account for the nonideal rejection. The actinyl ions may enter the nanochannels of GOM in a horizontal position, which could ease the steric constraint and lead to permeation of these actinyl ions through the GOM.

Based on the above discussions, we propose a schematic mechanism (Fig. 5) for ion sieving of actinides (actinyl ions) and lanthanides in the GOM. In brief, the nanochannels with a desired interlayer spacing generated from the stacking of GO sheets serve as the main pathways for the permeation of lanthanide ions, while the defects of GO, stacking voids in GOM, and non-uniform distribution of the

interlayer spacing of GOM cause un-desired permeation of actinyl ions. Nevertheless, the exact mechanism is still open to debate and it should involve the interplay of many chemical and physical processes. The findings from this work are expected to inspire more future theoretical and experimental efforts on these issues.

In summary, a novel strategy for An/Ln separation is developed through a facile coupling of chemical oxidation and GOM sieving and with the aid of solvent extraction. Efficient group separation of lanthanides from actinides under highly acidic conditions has been achieved with this strategy. We expect the coupling of multiple stages of sieving together would help further enhance the separation efficiency. Moreover, this separation strategy can be further modified and extended to fulfill other separation tasks in the nuclear fuel cycle, including the separation of actinides from a handful of key fission products (Sr, Cs, Zr, Pd, etc.) and the separation of Am from Cm. Relevant work is currently underway in our lab.

## Methods

### Chemicals and materials

All chemicals were of analytical grade or higher and used without further purification. Deionized water was used in all the experiments. $U_3O_8$ powder and $^{239}Pu(NO_3)_4$ solution were obtained from the stockpile in the Institute of Nuclear and New Energy Technology (INET), Tsinghua University. $^{237}NpO_2$ powder was obtained from the China Institute of Atomic Energy (CIAE). $^{241}AmO_2$ powder was purchased from Shenzhen Isotope Industrial International Co., Ltd. The actinide oxide powders were dissolved in nitric acid solution to obtain their respective stock solution. $Ln(NO_3)_3 \cdot 6H_2O$ (Ln = Ce, Nd, Eu, and Gd) salts in the analytical grade were purchased from Shanghai Aladdin Biochemical Technology Co., Ltd and dissolved respectively in the nitric acid solution for the permeation test. ***Caution! All the actinides utilized in this work are α-emitting radioisotopes and relevant experiments were performed in a radiological facility dedicated to studies on transuranic elements.***

### Synthesis of GO

Graphene oxide (GO) was prepared by the modified two steps Hummers' method[49,50]. In a typical synthesis protocol, 4.0 g of 200 mesh natural graphite oxide flakes (Alfa Aesar), 48 mL of concentrated sulfuric acid (Beijing Chemical Works), 8.0 g of potassium permanganate (Shanghai Aladdin Biochemical Technology Co., Ltd), and 8.0 g of phosphorus pentoxide (Beijing Chemical Works) were added in turn to synthesize pre-oxidized GO, which was then washed by deionized water till the vacuum filtrate becomes neutral. Then, 3.0 g of the pre-oxidized GO was dried in an oven at 60 °C for 12 h and was further oxidized with 3.0 g sodium nitrate (Beijing Chemical Works) and 18.2 g potassium permanganate (Shanghai Aladdin Biochemical Technology Co., Ltd.) in an ice bath. The mixed solution was then transferred to a 35 °C water bath for 2 h. After that, the solution was diluted with 300 mL of deionized water, followed by adding 10 mL of 30% hydrogen peroxide (Shanghai Aladdin Biochemical Technology Co., Ltd.) to completely react the excess potassium permanganate. The mixture was purified in dialysis tubes for a week, and followed by centrifugation at 8000 r/min twice to remove the liquid supernatant. The retained sticky GO dispersion was diluted to a suspension with 0.06 g/L GO.

### Preparation of GOM

GO membrane (GOM) was prepared by vacuum filtration of 10 mL of the 0.06 g/L GO suspension through a polyethylene terephthalate (PET) nuclear track membrane (5.0 cm in diameter and 12.5 μm in thickness) with an average pore size of 0.22 μm. After filtration, GOM was dried in the air at room temperature. The porous PET membrane acts as a mechanical support for the permeation test. The thickness of the GOM was about 150 ± 10 nm as estimated through SEM images.

### Oxidation of An/Ln Ions

The oxidation of lanthanide and actinide ions was performed by mixing the metal-containing 3.0 mol/L $HNO_3$ solution with $NaBiO_3$ solid powder vigorously for 2 h. The solid/liquid ratio was ~15 g/L. To confirm the successful oxidation of certain metal ions, the $NaBiO_3$ solid residue was removed by filtration before spectrometric characterization (U, Np, Pu, Am) or image acquisition (Ce) (Supplementary Fig. 5).

### Permeation test

An H-shaped glass filter composed of a feeding compartment (FC) and a receiving compartment (RC) was used in the permeation test. The solutions in both FC and RC were agitated by magnetic stirring during the permeation test. For Test I (corresponding to Fig. 3a–c), 10 mL of actinides/lanthanides solution and isovolumetric 3.0 mol/L $HNO_3$ solution were filled in FC and RC, respectively. For Test II (corresponding to Fig. 3d–f), the FC was filled with 10 mL of actinides/lanthanides solution and the RC was filled with 5 mL of 3.0 mol/L $HNO_3$ solution and 5 mL of 0.1 mol/L TODGA/n-dodecane solution. For Test III (corresponding to Fig. 3g–i), the FC was filled with the mixture of actinides/lanthanides solution and ~15 g/L $NaBiO_3$ solid powder, while the RC contained 5 mL of 3.0 mol/L $HNO_3$ solution and 5 mL of 0.1 M TODGA/n-dodecane solution. The initial concentrations (or radioactive count rates) of each ion in the three tests are: 2.7E-06 mol/L (~1000 α counts per minute (cpm)/mL) $^{237}Np$, 3.0E-07 mol/L (~10,000 α cpm/mL) $^{239}Pu$, 5.4E-09 mol/L (~10000 α cpm/mL) $^{241}Am$, 4.2E-03 mol/L (~1000 ppm) $^{238}U$, 7.1E-3 mol/L (~1000 ppm) Ce, 6.9E-03 mol/L (~1000 ppm) Nd, 6.6E-03 mol/L (~1000 ppm) Eu, and 6.4E-03 mol/L (~1000 ppm) Gd.

### Gamma irradiation

The γ-ray irradiation of GOM was carried out at room temperature. Typically, GOM was first soaked in 3.0 mol/L $HNO_3$ solution and the resulting wet membrane along with the nitric acid solution was directly irradiated using a $^{60}Co$ source at INET, and the dose rate was 120 ± 5 Gy/min. Three samples with cumulative doses of 10 kGy, 50 kGy, and 100 kGy were used for subsequent characterization and ion-sieving tests.

### Characterizations and measurement

Atomic force microscopic (AFM) images of graphene oxide were taken on an Asylum Research Cypher VRS scanning probe microscopy system (Oxford Instruments) and scanning electron microscopic (SEM) images were collected via a field-emission SEM (Merlin, Zeiss, Inc.) at high vacuum with an accelerating voltage of 5 kV. X-ray photoelectron spectroscopy (XPS) spectra for C1s and O1s were collected on the X-ray photoelectron spectrometer (ESCALAB Xi + , ThermoFisher Scientific) with a monochromatic Al X-ray radiation source operated at 15 kV and 10 mA. XRD spectra were collected on a X-ray diffractometer (Rigaku MiniFlex600, Cu Kα radiation) in the 2θ range from 3° to 15° with an interval of 0.01°. Raman spectra of the GO samples were collected using HORIBA Jobin Yvon's LabRAM HR Evolution Raman spectrometer with a 532 nm laser excitation at a power of 0.325 mW/cm². Fourier-transform infrared (FT-IR) spectroscopy was carried out by ThermoFisher Scientific Nicolet iZ10 spectrometer using the attenuated total reflection (ATR) method. Thermal decomposition of the freestanding GO membrane was carried out with a TA Instruments SDT-Q600 Simultaneous TGA/DSC, with a constant $N_2$ flow of 100 mL/min from room temperature to 900 °C at 5 °C/min. The countings of $^{237}Np$, $^{239}Pu$, and $^{241}Am$ were determined by Ultra Low Background Liquid Scintillation Detector (Quantulus 1220, PerkinElmer) and alpha spectrometry (Alpha Analyst 7200-12, Canberra). The concentrations of $^{238}U$, Ce, Nd, Eu, and Gd were measured by Inductively Coupled Plasma Atomic Emission Spectrometer (ICP-AES, ARCOS, Inc.). All the absorption spectra were collected on Cary 6000i UV−Vis−NIR spectrophotometer (Agilent, Inc.).

## Data availability

The authors declare that data relating to the characterization of materials, general methods, experimental procedures, and separation studies are available within the article and the Supplementary Information or from the corresponding authors upon request.

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

## Acknowledgements
We acknowledge the financial support from the Natural Science Foundation of Beijing Municipality (No. JQ20041 to C.X.) and the National Natural Science Foundation of China (Nos. 21790372 to C.X., 21976104 to Y.L., and 22006090 to Z.W.).

## Author contributions
Z.W., Y.L., and C.X. conceived the experiments. Z.W. and L.H. executed the experiments, collected and analyzed the data. X.D. assisted in the radiological experiments. T.W. and Q.Q. assisted in the GOM synthesis. J.C. participated in the results discussion. Y.L. and C.X. supervised the studies and wrote the manuscript.

## Competing interests
The authors declare no competing interests.
