## [Peer Review File · Nature Communications]

Ion sieving in graphene oxide membrane enables efficient actinides/lanthanides separationREVIEWER COMMENTS

Reviewer #1 (Remarks to the Author):

The manuscript presents results of experiments with graphene oxide membranes aimed on separation of actinides and lanthanides. The idea is to use strong oxidant and diffusion of oxidized ions across GO membrane with specific "nanochannel" size. Main result is that oxidized actinides diffuse a lot slower across the membrane compared to lanthanides. The results are of interest but the paper has some critical issues which do not allow adequate evaluation of results and suggested mechanism of separation. I list below several issues related to absence of information about experimental details and characterization of GO used to prepare membranes. These are all standard data. Authors provided adequate characterization of GO in their earlier study of differently oxidized GOMs in nitric acid (ref 21), not clear why this characterization is not provided in the new study. It is impossible to discuss permeation properties of GOMs e.g. without knowing degree of GO oxidation.

I also miss detailed discussion of chemistry involved in the studied systems. GO is chemically far from inert. There are many studies which report rather high sorption of actinides and lanthanides by GO. This literature is easy to find and needs to be cited and discussed. In principle, the sorption might modify permeation channels – either by expanding inter-layer distance or, in contrary by filling the channels and blocking solution pathways. It is not clear if adding HNO₃ solutions of each element separately and all together simultaneously will modify chemically GO in the membranes. No data related to membranes studied AFTER filtration experiments are presented, for example XRD of membranes after permeation tests is not presented.

I also have some doubts about hydration diameters used in the study and consider the size of GOM nanochannel calculation to be incorrect (even if it was used in some earlier high profile publications).

There are also more general questions related to proposed mechanism of separation in relation to simple concept of size related diffusion/blocking. For example, authors incorrectly assign diffusion of ions across GOMs to osmotic effect (see comment 7 below) . Another example, authors state that size of ions is too large compared to the size of permeation channels to explain high rejection. However, rejection is still not close to zero. 5% is still some significant permeation. How the size exclusion still allows diffusion but at slower rate? Discussion of (variety) of proposed in literature permeation mechanisms is absent in the paper. The real membrane structure is rather different compared to idealized picture shown in figure 2d, see for example this study: <https://doi.org/10.1021/acsnano.8b02015> or that one : <https://doi.org/10.1021/acsnano.8b07573>

My overall conclusion is following. I think that experimental observation of lower diffusion rates for oxidized actinides as compared to lanthanides justifies publication in Nature Comm. After all, if separation works, that is important result and complete explanation might come later. However, theoretical models are too speculative and at best can be only considered as hypothetic with many reservations and after mentioning other hypothetic possibilities. Also, the paper can not be advised to publication before detailed characterization of GO and GOMs are added and all experimental details required for possible reproducing of results are provided

See more details below.

1. Detailed characterization of GO used for preparation of GOMs is not presented. Authors need to provide XPS with C/O ratio, FTIR, TGA, flake size etc. It is difficult to discuss results of study without this basic information. The same about GOM preparation. Citing the text "In this work, a GOM with an interlayer spacing 90 (d) of 13.9 Å in 3.0 mol/L HNO₃ solution was precisely prepared (Detailed procedures are provided in the Supporting Information)."

I checked SI file and no procedures are enclosed. Authors need to provide it, otherwise their experiments are not reproducible. How was thickness of GO membrane measured? Image provided in Fig.2 has too low resolution to make any thickness measurements or to see something about texture of the membranes. Can better image of edge view of GOMs be provided? The value given is 115 nm, does it mean only one membrane was tested or the thickness is known with 1Å precision?

2. Detailed data for experimental procedures need to be added. Authors cited many ions tested for permeation but nothing about counter ions. What were exactly chemicals dissolved in HNO₃ is not found in the experimental sections. Experiments are impossible to reproduce without this

information. The section "chemicals" only reports info for 4 chemicals. There is no information about how solutions of Ce, Nd, Eu, Gd were prepared and from which chemicals, purity, source.

3. "The effective nanochannel size (μ) of this wet GOM is 10.5 Å ($\mu = d - e$, where $e \approx 3.4$ Å is the thickness of the monolayer GO nanosheet)." Layer separation of 3.4 Å is common for graphite. Graphene oxide layers are about 7-8 Å thick. Authors can verify this simple fact using AFM with single layered GO flake or check about 1000 references. It was indeed speculated in early GO membrane papers that interconnected non oxidized areas on GO form a network of "graphene capillaries". Evidence for this claim was never presented. The manuscript do not provide information about degree of oxidation of their GO in order to see if significant non oxidized regions could actually be present in their material. The standard GO is believed to be well oxidized with some not oxidized islands not connected to each other. (see e.g. DOI: 10.1002/adma.201000732) Authors themselves report that in absence of swelling the averaged interlayer distance of their GOMs is ~ 8.4 Å. There is no permeation of ions in absence of swelling. Therefore, correct method to calculate the size of nanochannel size in GOM subjected to swelling is 13.9 Å - 8.4 Å = 5.5 Å. The fact that ions penetrate across membrane with so small channels possibly indicates permeation of at least partly dehydrated ions. The whole concept of separation in this case needs to be revised. The size of hydrated ions provided in the paper most likely has no direct relation to permeation properties of GOMs. The diameter of "hydrated ions" can also be calculated in different ways. Authors selected specific method which fits into their expectations.

4. Authors assume that swelling in solutions of all used actinides and lanthanides remains to be the same as in pure nitric acid. That is also after adding strong oxidant. Obviously, adding additional ions which can be adsorbed by GOMs or chemically react with GOMs can modify swelling properties and size of nanochannels in many hardly predictable ways. Therefore, ideal is to verify swelling at least in solutions of each tested ions separately and test it for final mixture after adding oxidant. XRD testing of the GOMs after filtration experiments is needed to verify if the properties of membranes (and size of nanochannels) had remained the same or was modified by sorption of ions and possible chemical modification. I understand it can be difficult to record XRD for radioactive samples, but the possibility of changes needs to be discussed.

5. Permeation tests are performed with ions added in different concentration while the data are later plotted together using rejection ratio. The concentrations are provided in alfa counts. The driving force of diffusion is difference in molar concentration. Please provide molar concentrations for each experiment.

6. I would like to see more discussion of chemistry involved. For example, uranyl supposed to react with nitric acid thus forming uranyl nitrate. In this case hydration diameter calculation shown in SI file is not relevant, one need to consider hydration diameter of uranyl nitrate. See e.g. <https://doi.org/10.1021/j100123a037> What about nitrates of other oxide ions? Can authors provide relevant review of literature and discussion?

7. Authors incorrectly assign diffusion of ions across GOMs to osmotic effect. Citing the paper: "As shown in Fig. 3b and 3c, the smaller lanthanide ions (8.74 - 9.20 Å) in the FC could pass through the GOM ($\mu = 10.5$ Å) into the RC as a result of the osmosis effect." Permeation of ions in the setup used by authors is simple diffusion driven by concentration gradient. Adding extracting liquid helps to reduce concentration of ions on the permeate side and to maintain high gradient. Osmotic effect is related to flow of water from permeate side towards ions not capable to cross the membrane, thus diluting the feed solution. Authors cite in several places "osmosis conditions" of their experiments. But was the osmotic effect actually observed? I wonder if osmotic rise was observed on the feed side when the size of ions was too high for permeation across the membrane, please take pictures as a function of time if that is the case. Was the dilution of not penetrating ion solutions due to osmotic effect registered using any of used concentration measurement methods? If the osmotic rise is not observed, it might be indication of complete chemical blocking of permeation channels by adsorbed ions rather than sieving effect.

Reviewer #2 (Remarks to the Author):

“Ion Sieving in Graphene Oxide Membrane Enables Efficient Actinides/Lanthanides Separation”
Zhipeng Wang, Liqin Huang, Xue Dong, Tong Wu, Qi Qing, Yuexiang Lu, Jing Chen, and Chao Xu,
Nat. Commun., NCOMMS-22-30550

This manuscript discusses the application of graphene oxide membranes to achieve a lanthanide/actinide separation based on size discrimination between the larger, highly oxidized actinyl ions (AnO_{2n} , where n is +1 or +2) versus the smaller Ln^{3+} ions. To accomplish this, a semi-permeable membrane was formed by coating a porous nuclear track membrane with graphene oxide membranes. The separation was driven by osmosis from a concentrated solution to a solution free of ions, the smaller Ln^{3+} ions traversed the membrane more readily than the larger AnO_{2n} ions. To enhance the separation, a liquid-liquid extraction system was added to the receiving solution. The manuscript is generally well written, with no obvious grammatical issues. However, there are some issues, which need to be addressed before this manuscript can be recommended for publication. I recommend publishing after minor revisions.

Comments:

- 1) This is a very interesting approach and seems to have promise, however the kinetics of the diffusion across the membrane is very slow, taking at least 5 h to reach a 50% removal of the Ln^{3+} and over 24 h to get close 90%. This raises questions about the practicality of the approach.
- 2) Following the concerns with long incubation times, how resistant are the membrane to radiolytic degradation, the dose from actual used nuclear fuel solutions are very high?
- 3) The separation is not actually that efficient, in the best case there is still 5–10% of the Ln in the starting solution and 5–10% of the An in final solution.
- 4) What is the proposed mechanism for the movement of the actinyl ions from the starting solution to the end solution, there seems to be too high of a yield just based on diffusion through the membrane. Also, the waste stream (the receiving solution) has a relatively high actinide concentration, this does not seem conducive for a final disposition of the nuclear waste.
- 5) Finally, the authors give no comparison of their results to other methods trying to achieve a similar separation, this is needed. For example see *Ind. Eng. Chem. Res.* 2014, 53, 1624–1631 or *Inorg. Chem.* 2016, 55 (17), 8913-8919.

Reviewer #3 (Remarks to the Author):

Comments:

In this communication, authors have reported their research pertaining to the selective separation of lanthanides from actinides after oxidizing the later. They have used tailor made GO membrane as a support for the separation of metal ions facilitated by TODGA extractant. After reading the manuscript many times, I did not find any new breakthrough research in this article that qualifies its publication in “Nature Communication”. Off course, this is a nice piece of work and may be published as a regular full article in other relevant journal. My decision is based on following points.

-Preparation of GO with desired inter-layer spacing is not a new concept. Indeed, it has been prepared earlier also by the authors group and the same has been referred in this manuscript (ref-18 – 20).

-GO membrane was tested with lanthanides and actinides (in different oxidation state). What is the origin of the mixer of lanthanides and actinides (U, Pu, Np and Am) feed? If authors target is lanthanide/actinide separation, then it should be lanthanides and Am only? If their target is the separation of all lanthanides from a series of actinides, then they should also comment on the selectivity of other fission element and process chemicals, such as Cs, Sr, Fe, Pd etc, which I am sure, will pass through this GO membrane.

-Due to slow natural diffusion of metal ions within the pristine GO membrane, authors have used TODGA as the carrier extractant. Therefore, the overall separation becomes a carrier facilitated transport, and the selectivity amongst the metal ions will be decided by the carrier ligand, not by the membrane architecture. If it is so, then proposed methodology has no advantage of using specially designed GO membrane. Instead, it will be a poor choice for the researchers.

-The results with TODGA/GO membrane also shows about 8-10% permeation of Am, Np and Pu (dioxocations) along with lanthanides. Therefore, this level of contamination of in the lanthanide product can not be said as selective permeation.

-Separation factor of lanthanides over UO_2^{2+} are in the range of 100 - 400 (maximum). In fact, such a separation factor can also be obtained with TODGA supported polymeric membranes where TODGA itself has poor selectivity for UO_2^{2+} cation over Am and lanthanides (Dalton Trans., 2015, 44, 515-521, Solv. Extr. Ion Exch., 2001, 19, 91-103). This aspect should have been highlighted in the manuscript and reasoning should have been given about the advantages of using such a sophisticated GO membrane over the conventional polymeric membrane.

-The 90% transport of lanthanides took about 24 h in TODGA assisted GO membrane in this work. This can be achieved in less than 4 hrs if one uses a polymeric membrane support for TODGA. Authors have not highlighted this fact in their manuscript.

Overall, my impression is that this work does not has sufficient novelty and breakthrough research which can be published in Nature communication. However, undoubtedly, this is a nice piece of work, and can be published as a regular full article in other journal which authors find most suitable.

Reviewer #4 (Remarks to the Author):

In this report the authors report using a graphene oxide layer to effectively sieve actinide dioxocations from lanthanide spherical cations. The authors control graphene oxide swelling to an optimal channel size $\approx 10.5 \text{ \AA}$ while adding a chemical driver, solvent extraction, to allow the smaller lanthanide spherical cations to pass through while the larger actinyl cations are held back in the original feed solution. Graphene oxides have been used in the past to purify water and used to remove heavy metals from solution. This work represents a very clever and relatively easy approach to achieving one of the most challenging separations known to man. It is unclear to this reviewer at this time if this process can be scaled up to process tons of used fuel, but at the very least it can be employed as an analytical technique for preconcentration of lanthanides from mixed matrices. The authors present a well written manuscript with conclusions that are consistent with the data presented. This reviewer recommends publishing but has a few suggestions/questions for the authors to address before publishing.

#1 In the supplemental information the authors show a picture of the experimental apparatus in use with the highly insoluble sodium bismuthate in the FC solution. Did the insoluble sodium bismuthate collect on the GOM surface (on the FC side) and clog the channels that allow the lanthanides pass through? If not, do the authors anticipate this if the experiments are run longer than 24 hours? Do the authors anticipate this to be problematic if the process is scaled up to accommodate more concentrated actinide solutions and thus more sodium bismuthate?

#2 The authors state that based on figure 4, with mixed waste, the permeation of the lanthanides reaches >90% and the permeation ratios of actinides are retained at <9%. This implies all actinides behaved the same. They define the separation factor and present the highest separation factor of 400 for U/Ce. Although this conclusion is drawn directly from the results, it is a little misleading since the most important separation factors gained from this study should involve the actinides that require oxidation state control under these conditions, i.e., Np, Am and Pu. Under these conditions, hexavalent uranium will be the most stable actinyl ion. From the 3D bar graph it is hard to determine the differences between R(%) for the actinides. This reviewer suggest the authors provide a more detailed description of the R(%) for each actinide (e.g. alpha counts or concentrations from the LSC counting for the FC and RC solutions for all actinides). Provide a hypothesis to explain why Am, Np and Pu concentrations are increasing in the RC solutions to

account for the lower separation factors for all lanthanides compared to U. Are Np, Am, and Pu being reduced back to spherical cations during the 24h permeation period? If so, how can that happen in the presence of excess sodium bismuthate? Are Np, Am and Pu being reduced by the graphene oxide layer upon contact?

#3 It appears from the 3D bar graph that Eu (II) and Nd(III) are the least permeated lanthanides during the mixed waste test. This could help account for the lowest reported separation factor $Nd/Am = 126$. Can the authors explain the reduced permeation for Eu and Nd in the mixed waste test?

#4 How robust against alpha radiation are the graphene oxide layers? Can more concentrated actinide solutions with increased alpha decay events destroy the channels or alter the channel size in the graphene oxide layer to reduce efficiency of the separation?

The reviewer feels the manuscript can be greatly strengthened by addressing these suggestions/questions.

Response to reviewers' comments

GENERAL: We are grateful to the reviewers for all their valuable comments, which help to improve the manuscript significantly. We have considered all the suggestions and criticisms carefully and have accordingly revised the manuscript. The revisions are marked in an additional version of the manuscript to facilitate reviewing. The detailed responses are listed below, with the original comments showing in italics. We hope the reviewers agree that the revised manuscript can fulfill the requirements for publication in *Nat. Commun.*

Reviewer #1:

The manuscript presents results of experiments with graphene oxide membranes aimed on separation of actinides and lanthanides. The idea is to use strong oxidant and diffusion of oxidized ions across GO membrane with specific "nanochannel" size. Main result is that oxidized actinides diffuse a lot slower across the membrane compared to lanthanides. The results are of interest but the paper has some critical issues which do not allow adequate evaluation of results and suggested mechanism of separation. I list below several issues related to absence of information about experimental details and characterization of GO used to prepare membranes. These are all standard data. Authors provided adequate characterization of GO in their earlier study of differently oxidized GOMs in nitric acid (ref 21), not clear why this characterization is not provided in the new study. It is impossible to discuss permeation properties of GOMs e.g. without knowing degree of GO oxidation.

Response: We appreciate greatly the reviewer's constructive comments. Regarding the issues of absence of information about experimental details and characterization of GO used to prepare membranes, we have provided more experimental details for the preparation of the GO and GO membrane (GOM) and have also conducted additional experiments to characterize the GO/GOM to explicate these issues. Details have been presented in the responses to comments 1 and 2.

I also miss detailed discussion of chemistry involved in the studied systems. GO is chemically far from inert. There are many studies which report rather high sorption of actinides and lanthanides by GO. This literature is easy to find and needs to be cited and discussed. In principle, the sorption might modify permeation channels – either by expanding inter-layer distance or, in contrary by filling the channels and blocking solution pathways. It is not clear if adding HNO₃ solutions of each element separately and all together simultaneously will modify chemically GO in the membranes. No data related to membranes studied AFTER filtration experiments are presented, for example XRD of membranes after permeation tests is not presented.

Response: We appreciate the reviewer's kind reminder. The reviewer is absolutely correct, GO is not chemically inert and has been widely employed as adsorbents for actinides and lanthanides in many reports (*Phys. Chem. Chem. Phys.*, 2013, 15, 2321-2327; *Chem. Eng. J.*, 2012, 210, 539-546; *Environ. Sci. Technol.*, 2012, 46, 11, 6020-6027; *Inorg. Chem. Front.*, 2015,

2, 593-612; *ACS Appl. Mater. Interfaces*, 2020, 12, 45122-45135; etc.). We have cited several relevant references in our revised manuscript. To reveal the adsorption behavior of the GO synthesized in our work, we have also conducted adsorption experiments for representative actinides and lanthanides under a variety of experimental conditions. The adsorption results are shown in **Figure R1** and indicate the following facts: 1) The adsorption ability to U(VI), Am(III) and Eu(III) by the GO increases with the increase of pH of the aqueous solution. Appreciable adsorption was only observed at relatively high pH conditions. This observation is well consistent with the findings in previous reports (e.g., *Phys. Chem. Chem. Phys.*, 2013, 15, 2321-2327). 2) The adsorption ability to Am(III) and Eu(III) is stronger than that to U(VI) under the same conditions. This trend has also been reported in the literature (same ref as above). 3) The adsorption of U(VI), Am(III) and Eu(III) from highly acidic solutions (1.0 and 3.0 M HNO₃) by the GO is quite weak. In particular, the adsorption of metal ions from 3.0 M HNO₃ by the GO is close to the detection limit. The above facts suggest that the adsorption by the GO unlikely plays a critical role in the permeation of the actinides and lanthanides under the sieving conditions (3 M HNO₃) in our present work. These results will be collectively discussed later with other additional experimental results to reveal the permeation mechanism (see response to comment 3).

Figure R1. Adsorption of ²⁴¹Am(III), ^{152,154}Eu(III) and U(VI) by the GO under different aqueous acidities. Initial concentrations of ²⁴¹Am(III), ^{152,154}Eu(III) and U(VI): ~10000 α cpm/mL, ~10000 β cpm/mL and 4 mM, respectively. Solid (GO)/liquid ratio: 1 mg GO/ 10 mL aqueous solution. Adsorption time: 24 h.

Moreover, we have characterized the GOM after filtration experiments by XRD. As shown in **Figure R2**, the XRD patterns before and after permeation show no obvious differences, suggesting the interlayer spacing of the GOM has not been affected by the permeation of the metal ions. Under the permeation conditions in our work, we expect the interlayer spacing of the GOM is mainly determined by the high concentrations of nitric acid. These results have also been discussed in the responses to comment 4.

Figure R2. XRD patterns of GOM before and after metal ion permeation for 24 h. The feed solution contains (a) 5 mM U(VI); (b) 5 mM Nd(III); (c) 18 mM Ce(III) + 29 mM Nd(III) + 1 mM Eu(III) + 1 mM Gd(III) (Corresponding to the composition of Ln ions in the simulated feed solution in **Table S2**), $[\text{HNO}_3] = 3.0 \text{ M}$.

I also have some doubts about hydration diameters used in the study and consider the size of GOM nanochannel calculation to be incorrect (even if it was used in some earlier high profile publications).

Response: We appreciate greatly the reviewer's comment. First, we have consulted more references on the hydration issue of actinide and lanthanide ions, and found out a more reasonable approach to calculate the hydration size of actinyl ions in our revised manuscript. Regarding the GOM nanochannel size calculation, we have also researched more literatures and conducted additional experiments, and then came up with an overall assessment on the validity and applicability of the calculation approach we used in this work. Details have been presented in the response to comments 3.

There are also more general questions related to proposed mechanism of separation in relation to simple concept of size related diffusion/blocking. For example, authors incorrectly assign diffusion of ions across GOMs to osmotic effect (see comment 7 below). Another example, authors state that size of ions is too large compared to the size of permeation channels to explain high rejection. However, rejection is still not close to zero. 5% is still some significant permeation. How the size exclusion still allows diffusion but at slower rate? Discussion of (variety) of proposed in literature permeation mechanisms is absent in the paper. The real membrane structure is rather different compared to idealized picture shown in figure 2d, see for example this study: <https://doi.org/10.1021/acsnano.8b02015> or that one: <https://doi.org/10.1021/acsnano.8b07573>

Response: Thanks for pointing out our mistake of assigning diffusion of ions across GOMs to osmotic effect. We have corrected this mistake in the revised manuscript. We have also consulted more references, especially those on GOM structure and ion sieving mechanism, and presented more discussions on the permeation mechanism in the revised manuscript. More details have been presented in the responses to comments 3 and 7.

My overall conclusion is following. I think that experimental observation of lower diffusion rates for oxidized actinides as compared to lanthanides justifies publication in Nature Comm. After all, if separation works, that is important result and complete explanation might come later. However, theoretical models are too speculative and at best can be only considered as hypothetic with many reservations and after mentioning other hypothetic possibilities. Also, the paper can not be advised to publication before detailed characterization of GO and GOMs are added and all experimental details required for possible reproducing of results are provided. See more details below.

Response: We appreciate the reviewer's comment, especially for recognizing the interesting An/Ln separation results afforded by the GOM sieving method. In our revised manuscript, we have provided more experimental details to ensure the reproducibility of our results by other researchers, and we have also substantially enriched the characterization of GO/GOM as well as the discussion on the separation mechanism to improve the theoretical insights of this work.

Comment 1. *Detailed characterization of GO used for preparation of GOMs is not presented. Authors need to provide XPS with C/O ratio, FTIR, TGA, flake size etc. It is difficult to discuss results of study without this basic information. The same about GOM preparation. Citing the text "In this work, a GOM with an interlayer spacing 90 (d) of 13.9 Å in 3.0 mol/L HNO₃ solution was precisely prepared (Detailed procedures are provided in the Supporting Information)." I checked SI file and no procedures are enclosed. Authors need to provide it, otherwise their experiments are not reproducible. How was thickness of GO membrane measured? Image provided in Fig.2 has too low resolution to make any thickness measurements or to see something about texture of the membranes. Can better image of edge view of GOMs be provided? The value given is 115 nm, does it mean only one membrane was tested or the thickness is known with 1 Å precision?*

Response: Thanks for raising this important issue. First, the detailed procedures for the preparation of GOM have now been presented in the Method section in our revised manuscript. Moreover, we have provided more detailed characterization (AFM, SEM, XPS, TGA, XRD, Raman and IR) of the prepared GO/GOM in our revised manuscript as suggested by the reviewer. Some of the characterization results (AFM, SEM, XPS and XRD) are now presented in the updated **Figure 2** in the main manuscript, while others (TGA, Raman and IR) are summarized in the Supporting Information. These characterization results do provide useful information to help us reveal the relevant permeation mechanism. The 115 nm thickness in our original manuscript was roughly calculated from the less-enlarged SEM image and we expect this value is associated with a large uncertainty. In our revised manuscript, we have collected more enlarged SEM image of the edge view of several GOMs and determined the thickness of the GOM as 150 ± 10 nm.

Comment 2. *Detailed data for experimental procedures need to be added. Authors cited many ions tested for permeation but nothing about counter ions. What were exactly chemicals dissolved in HNO₃ is not found in the experimental sections. Experiments are impossible to*

reproduce without this information. The section “chemicals” only reports info for 4 chemicals. There is no information about how solutions of Ce, Nd, Eu, Gd were prepared and from which chemicals, purity, source.

Response: We appreciate the reviewer’s kind reminder. Detailed experimental procedures have been added in the revised manuscript. In brief, all the starting metal ions (U, Np, Pu, Am, Ce, Nd, Eu, Gd) for the permeation tests are nitrate salts in nitric acid solutions. Specifically, the stock solutions of U(VI), Np(V) and Am(III) were prepared by dissolving their respective oxide (U_3O_8 , NpO_2 , AmO_2) in nitric acid solution. Pu(IV) is a long standing solution of ~ 5 mM $Pu(NO_3)_4$ in 1 M HNO_3 . A desired volume of these stock solutions was taken and added into the Feeding Compartment for the permeation tests. All the stock solutions for lanthanides are in the form of 0.1 M $Ln(NO_3)_3$ in 1 M HNO_3 .

Comment 3. *The effective nanochannel size (μ) of this wet GOM is 10.5 \AA ($\mu = d - e$, where $e \approx 3.4 \text{ \AA}$ is the thickness of the monolayer GO nanosheet).” Layer separation of 3.4 \AA is common for graphite. Graphene oxide layers are about $7-8 \text{ \AA}$ thick. Authors can verify this simple fact using AFM with single layered GO flake or check about 1000 references. It was indeed speculated in early GO membrane papers that interconnected non oxidized areas on GO form a network of “graphene capillaries”. Evidence for this claim was never presented. The manuscript do not provide information about degree of oxidation of their GO in order to see if significant non oxidized regions could actually be present in their material. The standard GO is believed to be well oxidized with some not oxidized islands not connected to each other. (see e.g. DOI: 10.1002/adma.201000732)*

Authors themselves report that in absence of swelling the averaged interlayer distance of their GOMs is $\sim 8.4 \text{ \AA}$. There is no permeation of ions in absence of swelling. Therefore, correct method to calculate the size of nanochannel size in GOM subjected to swelling is $13.9 \text{ \AA} - 8.4 \text{ \AA} = 5.5 \text{ \AA}$. The fact that ions penetrate across membrane with so small channels possibly indicates permeation of at least partly dehydrated ions. The whole concept of separation in this case needs to be revised. The size of hydrated ions provided in the paper most likely has no direct relation to permeation properties of GOMs. The diameter of “hydrated ions” can also be calculated in different ways. Authors selected specific method which fits into their expectations.

Response: We MUST thank the reviewer’s critical and insightful comments that lead us to an in-depth thinking of the underlying chemical force for the sieving. We have taken these comments very seriously and made great effort to find the most reasonable mechanism to explain the sieving behavior of the GOM in our work.

(1) The issue on effective nanochannel size of GOM.

First of all, we totally agree with the reviewer that the monolayer thickness is about 3.4 \AA for graphite and $\sim 7-8 \text{ \AA}$ for graphene oxide, which has been confirmed in numerous reports. Moreover, as pointed out by the reviewer, the surface of certain GOs has been well visualized in atomic scale through aberration corrected TEM (acTEM) characterization (*Adv. Mater.*, 2010, 22, 4467-4472; *Phys. Chem. Chem. Phys.*, 2022, 24, 2318-2331), which suggests that GO contains mainly three different regions, the pristine (non-oxidized) graphene region, the disordered oxidized region, and the defective (pinhole) region. The relative portion of these

regions is expected to be strongly dependent on the preparation and post-processing (such as reduction or thermal annealing) procedures. Erickson et al. did suggest that there are only small non-oxidized and defective islands in the GO and these islands are not connected to each other (*Adv. Mater.*, 2010, 22, 4467-4472). In this specific case, these non-oxidized regions on GO are unlikely able to form a network of “graphene capillaries” in the GOM for metal ion permeation. Nevertheless, we must point out that the surface structure of GO should depend on detailed synthesis protocols and can be further modified in different ways. For example, the pristine graphitic domains on GO could be enhanced and connected through ways such as thermal annealing (*Mat. Today*, 2021, 50, 44-54). For the GOM in this work, we have deliberately selected a synthesis protocol that could lead to low oxidation degree according to the results of our previous work (*Sci. Adv.*, 2021, 8, 2002717). Unfortunately, we currently have no evidence from acTEM to illustrate and differentiate the exact surface domains on the GO sheets. However, we have tried to understand the surface structural features based on the results from other characterizations.

First, XPS analysis (**Figure 2e** in our revised manuscript) suggests that the C/O ratio of GO in our work is ~ 2.45 . This value is quite large as compared to common GOs in literature (*J. Phys. Chem. C*, 2017, 121, 20489-20497; *Mat. Today*, 2021, 50, 44-54) and indicates the oxidation degree of our GO is relatively small. The C/O ratio of GO in our work is even higher than that of the GO with apparent connected graphitic domains evidenced from TEM analysis (*Mat. Today*, 2021, 50, 44-54). In this case, it is reasonable to assume relatively large and connected graphitic domains exist in our GO and stacking of these GO sheets together may generate graphitic channels to allow ion permeation. Moreover, Raman analysis (Figure S2 in our revised Supporting Information) of our GO gives an I_D/I_G ratio at ~ 1.1 . This relatively high I_D/I_G ratio is not necessarily related to the oxidation degree but it suggests there is a high density of vacancies and defects in the GO (*J. Phys. Chem. C*, 2017, 121, 20489-20497; *J. Phys. Chem. C*, 2015, 119, 10123-10129). These structural information from XPS and Raman will be further discussed later in combination with the issue of metal hydration to explain the permeation behavior in the GOM.

(2) The issue on size of hydrated lanthanide and actinide ions.

For the diameters of hydrated spheric metal ions in aqueous solution, there are not too much disputes and the values are mainly obtained through measuring the conductivity/diffusion rate or electromigration velocities and then being calculated according to the Stokes' law (*J. Phys. Chem.*, 1959, 63, 1381-1387; *Acta Chem. Scand. A*, 1981, 35, 653-661; *Sep. Purif. Technol.*, 2012, 86, 119-126.). Generally, the diameter of hydrated trivalent lanthanides is in the range of 9.0 – 9.3 Å, which indicates that the boundary of these hydrated ions stretches apparently beyond the first hydration shell of the lanthanides. Moreover, the diameter of hydrated trivalent lanthanides exhibits a monotonically increasing trend, which is opposite to the trend of lanthanide contraction effect (in similar coordination environment). This is because the higher charge density of heavier lanthanide make it attract more outer-coordinated water molecules in the second hydration shell.

However, the exact size or diameters of the hydrated actinyl ions (with contributions from outer-sphere waters) has been rarely investigated in the same manner as the lanthanides in literatures, probably due to the inapplicability of the characterization method or model for the actinyl ions with a unique linear configuration. Therefore, we tried to estimate the hydration

size of actinyl ions by ourselves in our original manuscript. We carefully revisited our original estimation method and found it is not scientifically sound. Herein, to estimate the hydration size of actinyl ions more accurately, we have consulted more literatures dealing with the bulk hydration issue of actinyl ions in aqueous solution (*J. Am. Chem. Soc.*, 2005, 127, 14250-14256; *J. Phys. Chem. B*, 2014, 118, 14373-14381; *J. Phys. Chem. A*, 2013, 117, 6421-6432; *J. Chem. Phys.*, 2016, 145, 224502). In general, for hydrated uranyl ion, there is an apparent asymmetry around the central metal, because the axial regions are quite different from the equatorial region. The very first hydration shell is around the equatorial plane and consists of approximately five water molecules, which has also been experimentally confirmed. These water molecules undergo strong polarization effects and partial charge transfer from their direct interaction with the metal cation. The second hydration shell includes water molecules belonging to the bulk solvent. Molecular dynamics (MD) results suggest that the overall hydration structure of UO_2^{2+} is strongly anisotropic (**Figure R3**), coupling of a conventional hydration sphere in the equatorial region with clathrate-like caps around the axial region. As shown in **Figure R3**, the second hydration shell of uranyl ion obviously stretches further along the axial direction due to the presence of two yl-O atoms. Notably, the distance of shell boundary from U in the axial direction is about 1.2 times longer than that in the equatorial direction. If we assume the ionic diameter along the equatorial direction is similar to that of trivalent lanthanide ions (this is reasonable because the U-O bond distance in the equatorial plane is close to that of Ln-O bond distance in similar coordination environment), then the overall size of the uranyl ion will be about 1.2 times larger than that of the lanthanide ions. Consequently, the diameter of the uranyl ion can be estimated to be $>10.8 \text{ \AA}$ (The ionic size of U(III) and Ce(III) is similar and the hydrated diameter of Ce(III) is 9.02 \AA). For other actinyl ions such as Np(VI), Pu(VI), and Am(VI), we believe the sizes of their hydrated ions are slightly larger than U(VI) if there is a similar trend in hydrated ion size across the actinyl series as that across the lanthanide series. The difference in An-O_{yl} bond length for the actinyl ion from U to Am is generally within 0.04 \AA (*Dalton Trans.*, 2017, 46, 2542-2550; *Inorg. Chem.*, 2019, 58, 3425-3434), suggesting that the overall diameter difference between the four hydrated actinyl ions might be less than 0.1 \AA . In our revised manuscript, we have corrected the size of hydrated actinyl ions accordingly and only suggest their diameters are approximately $> 10.8 \text{ \AA}$. It should be noted that although the corrected ionic size for actinyl ions is significantly smaller than that we originally estimated, their size is still larger than the size of nanochannel in GOM. (Note: Another way to estimate the size of hydrated actinyl ions is to roughly add the length of O=An=O to the corresponding hydrated spherical ions. This will result in a much larger size of the actinyl ions, i.e., $\sim 3.4 \text{ \AA}$ larger than the spherical ions in diameter. However, this way is not that reasonable because the hydration behavior of the yl-O atoms in the actinyl ion is very different than that of the central metal.)

Editorial Note: Reprinted from Pérez-Conesa, et al., "A hydrated ion model of $[\text{UO}_2]^{2+}$ in water: Structure, dynamics, and spectroscopy from classical molecular dynamics", *J. Chem. Phys.* 145, 224502 (2016) <https://doi.org/10.1063/1.4971432>, with the permission of AIP Publishing

Figure R3. MD results for the dynamical model of hydrated uranyl ion. (Copied from *J. Chem. Phys.*, 2016, 145, 224502)

Here, we must point out that the above discussions are based on the assumption that all the metal ions exist in hydrated forms in the solution, but have not considered the complexation of the metals with ligands other than water molecules. In fact, our permeation tests were conducted in 3.0 M nitric acid solution, in which the high concentration of nitrate ion may complex with the lanthanide and actinide ions and thus likely increase their overall ionic size during the permeation process. We have present detailed discussions on this issue in our following response to comment 6.

(3) The interplay between GOM structure and metal hydration.

The above discussions suggest hydrated lanthanides possibly permeate through the GOM *via* the nanochannel formed between pristine graphitic domains. However, we cannot exclude other possibilities. There must be an interplay between GOM structure and metal hydration, and quite a few questions remain. For example, do the metal ions undergo dehydration during the permeation process? Could the narrower channel formed between the oxidized domains of GO allow the permeation of lanthanide ions? Does the charge of the metal ions also play a role? These questions inspired us to further probe the permeation mechanism by revisiting our previous data and conducting additional experiments.

First, we revisited the metal permeation data through GOMs of different oxidation degrees in our previous report. We found that the oxidation degree has little influence on the permeation of K^+ ions (**Figure R4**). Notably, the GOM with the highest oxidation degree has an interlayer distance of ~ 8.0 Å under dry conditions and the distance swells to 11.4 Å when soaked in 3.0 M HNO_3 . In this case, the nanochannel size can be calculated to be 3.4 Å between oxidized regions and ~ 8.0 Å between non-oxidized regions (Note: even this GOM has the highest oxidation degree among four GOMs in that work, the C/O ratio of the GO is still >2 and thus non-oxidized regions may be connected on the surface). If we assume the K^+ ions permeate through this GOM through the oxidized channel (3.4 Å), most of the water molecules in the first hydration shell of K^+ must be dehydrated because the K-O bond of the hydrated ion is > 2.5 Å (*Inorg. Chem.*, 2012, 51, 425-438). Considering the similar permeation rate of K^+ ions through the four GOMs, we speculate that K^+ ions permeate through the large non-oxidized channels other than the narrower oxidized channels.

Figure R4. Layer spacing and metal permeation results for four GOMs with different oxidation degrees. (Copied from *Adv. Sci.* 2021, 8, 2002717.)

We have also performed additional experiments to see if the metal ions are hydrated or dehydrated when permeate through the GOM. Specifically, we performed permeation tests for two series of metal ions. One is the alkali metals Li^+ , Na^+ , K^+ , Rb^+ and Cs^+ , and the other is the alkaline earth metals Mg^{2+} , Ca^{2+} , Sr^{2+} and Ba^{2+} . We selected these two series of metal ions because the metal ions within each series possess the same charge but have different ionic sizes. Especially, the trend of the size change of the dehydration ions (also first-shell hydrated ions) is different from that of the size change of the so-called hydrated ions. For example, from Li^+ to Cs^+ , the ionic radius of the ions in crystals with the same coordination number increases from 0.60 Å to 1.69 Å, while the radius of their hydrated ions decreases from 3.82 Å to 3.29 Å (*J. Phys. Chem.*, 1959, 63, 1381-1387). If the permeation through GOM is based on size selection, then the permeation trend could provide information on the hydration status of the metal ions. The results for the two tests are shown in **Figure R5**. One can see that the permeation ratios more likely follow the size trend of the hydrated ions, i.e., smaller hydrated ion is associated with slightly higher permeation ratio. These results suggest the alkali and alkaline earth metal ions may pass through the GOM in their hydrated form. And if this is correct, the metal ions should pass through the large non-oxidized channels other than the narrower oxidized channels. Of course, we must point out that the permeation of lanthanides and actinides ions does not necessarily follow the same mechanism due to the big difference in their chemical properties from the alkali and alkaline earth metals.

Figure R5. Permeation of (a) alkali and (b) alkaline earth metal ions through GOM. Initial concentration of each metal: 5 mM, $[\text{HNO}_3] = 3.0 \text{ M}$.

(4) Possibility of other permeation mechanisms.

The above discussions suggest the ion sieving in our work is likely *via* the permeation of the hydrated lanthanide ions through the non-oxidized channels in GOM. However, we believe other permeation mechanisms might be also available, because the permeation results show that hydrated ionic size alone cannot fully explain the rejection characteristics. As pointed out by the reviewer, the rejection of actinyl ions is still not close to zero and 5% is still some significant permeation. This non-ideal rejection of actinyl ions might be related to the intrinsic structural feature of the GOM as well as the unique linear configuration of the actinyl ions. First, as reminded by the reviewer and mentioned by ourselves earlier, GO sheets usually have defects on the surface and the pinhole defects (usually with a diameter of several Å to tens of Å) could provide additional pathways for metal ion permeation. Moreover, the stacking of GO sheets is usually imperfect and stacking voids exist inside the GOM (*ACS Nano*, 2018, 12, 7855-7865), which also provide additional pathways for metal ion permeation. These pinhole defects and stacking voids in GOM may allow larger metal ions to pass through, resulting in some significant permeation of the actinyl ions. Furthermore, based on our current experimental results, we cannot totally exclude the possibility of dehydration of the metal ions inside the GOM. If some of the actinyl ions do undergo dehydration in the GOM channel, then this will also result in their more permeation. Additionally, the linear configuration of the actinyl ions is another factor need to be considered. The actinyl ions can either take a vertical or horizontal position relative to the GO sheet to pass through the nanochannels. The resisting force in the channel would be quite different for the two positions. If the actinyl ion enters the nanochannels of GOM in a horizontal position, the steric constraint will be eased and this may lead to permeation of these actinyl ions through the GOM. Also, the negatively charged yl-O atoms may undergo attraction or repulsion interactions with the functional groups of GO, bringing further complexity into the permeation mechanism. Another thing we would like to point out here is there might be another type of nanochannel generated from the stacking of GO sheets, because the distribution of the non-oxidized domains on GO is apparently not uniform for each GO sheet. Therefore, apart from the overlap of two oxidized domains or two non-oxidized domains, the overlap of one oxidized domain and one non-oxidized domain is also highly possible and would generate nanochannels with size between the other two types of channels. All these are important issues that have not gotten sufficient attention in the literature yet and require further theoretical and experimental investigations.

In addition, there are reports dealing with charge-selective sieving of metal ions using membranes (*J. Mater. Chem. A*, 2017, 5, 8051-8061). Since in our work the apparent charges of lanthanide ions (+3/4) and actinyl ions (+2) are different, we have conducted additional experiments to see if the ion charge also plays an important role in the metal ions' permeation through the GOM. We selected two metals for this investigation. One is Ce and the other is Np. The permeation behaviors of the Ce(III)/Ce(IV) and Np(V)/Np(VI) couples were compared and the results are shown in **Figure R6**. Obviously, the different charges on the metal show no appreciable influence on the permeation. The slightly lower permeation ratio of Ce(IV) as compared to Ce(III) should be caused by the strong complexation of nitrate with Ce(IV) in the solution.

Figure R6. Permeation of (a) Ce(III)/Ce(IV) and (b) Np(V)/Np(VI) through GOM. Initial concentration of metals: 6 mM Ce, ~1,000 α counts per minute (cpm)/mL for Np, $[\text{HNO}_3] = 3.0$ M. For Np(VI) permeation, NaBiO_3 was present in the feed solution to stabilize Np(VI). There was no TODGA extraction in the receiving compartment.

As seen from the above discussions, the real transport behavior of lanthanide and actinide ions in GOM in our work might be very complex and the sieving result should be an overall outcome from the interplay of size selection and various chemical interactions, with the former factor likely playing the most important role. Accordingly, we have substantially revised the discussions on permeation mechanism in our revised manuscript. Also, the scheme for ion sieving of actinides (actinyl ions) and lanthanides through the GOM has been revised accordingly.

Comment 4. *Authors assume that swelling in solutions of all used actinides and lanthanides remains to be the same as in pure nitric acid. That is also after adding strong oxidant. Obviously, adding additional ions which can be adsorbed by GOMs or chemically react with GOMs can modify swelling properties and size of nanochannels in many hardly predictable ways. Therefore, ideal is to verify swelling at least in solutions of each tested ions separately and test it for final mixture after adding oxidant. XRD testing of the GOMs after filtration experiments is needed to verify if the properties of membranes (and size of nanochannels) had remained the same or was modified by sorption of ions and possible chemical modification. I understand it can be difficult to record XRD for radioactive samples, but the possibility of changes needs to be discussed.*

Response: We appreciate greatly the reviewer's suggestion. We have accordingly performed XRD tests for the GOMs before and after filtration experiments. The XRD spectra for GOMs before and after permeation of given metal ions have been presented and discussed in our response to earlier comment and the results in **Figure R2** suggest the interlayer spacing of the GOM has not been affected by the permeation of the metal ions. The stability of the interlayer spacing is also consistent with our observation that neither lanthanide or actinide ions will be adsorbed by GO under highly acidic conditions and the spacing is mainly controlled by the acid. Moreover, we also collected XRD spectra before and after ion sieving in the presence of NaBiO_3 . As can be seen from **Figure R7**, the constant presence of strong oxidizing NaBiO_3 also has negligible influence on the interlayer spacing, further proving the robustness of the GOM in our work.

Figure R7. XRD patterns of GOM before and after sieving for 24 h in the presence of NaBiO₃. The initial feed solution is 3.0 M HNO₃ solution contains ~15 g/L NaBiO₃ solid powder.

Comment 5. *Permeation tests are performed with ions added in different concentration while the data are later plotted together using rejection ratio. The concentrations are provided in alfa counts. The driving force of diffusion is difference in molar concentration. Please provide molar concentrations for each experiment.*

Response: We appreciate the reviewer's kind reminder. We have provided molar concentrations of metal ions for each experiment in our revised manuscript.

Comment 6. *I would like to see more discussion of chemistry involved. For example, uranyl supposed to react with nitric acid thus forming uranyl nitrate. In this case hydration diameter calculation shown in SI file is not relevant, one need to consider hydration diameter of uranyl nitrate. See e.g. <https://doi.org/10.1021/j100123a037> What about nitrates of other oxide ions? Can authors provide relevant review of literature and discussion?*

Response: We fully agree with the reviewer on this point. Indeed, the uranyl ion would form complexes with nitrate ions in nitric acid solution and this issue has been investigated in a number of literatures (*J. Chem. Thermodyn.*, 2008, 40, 1001-1006; *J. Solution Chem.*, 2007, 36, 1093-1102; *J. Phys. Chem.*, 1993, 97, 5685-5692, etc.). In a normal nitric acid solution, uranyl can only form 1:1 complex ([UO₂(NO₃)⁺]) with nitrate ion, and the corresponding stability constant ($\log\beta$) has been determined by spectroscopic titration method. Based on a recently reported $\log\beta$ value of -0.62, we can calculate the speciation of U(VI) in nitric acid solution and the results suggest that ~40% of U(VI) ions are complexed with nitrate ion while most (60%) of U(VI) ions are in the free form for ~4 mM U(VI) in 3 M HNO₃ (the starting conditions for permeation test in our work). There are no reliable data for the stability constants for the complexes of Np(VI), Pu(VI) and Am(VI) with nitrate ion in aqueous solution. But considering the chemical similarity of these hexavalent actinyl ions to U(VI), it is reasonable to assume their stability constants are at the same level as that of U(VI) and thus an appreciable part of Np(VI), Pu(VI) and Am(VI) may also exist as 1:1 nitrate complex in the solution. Interesting, the less charged Np(V) could also form 1:1 complex with nitrate ion in aqueous solution and the complexation strength is comparable to U(VI) (*Radiochim. Acta*, 2010, 98, 71-75).

The stability constants for the complexation of Am(III) and some of the lanthanides (such as Nd(III)) with nitrate ion are also available in literatures (*Inorg. Chem.*, 2009, 48, 964-970; *Dalton Trans.*, 2014, 43, 14565-14569), thus similar calculations can be done to reveal the speciation of trivalent actinides and lanthanides in nitric acid solution under our experimental conditions. In general, the complexation of trivalent actinide and lanthanide ions with nitrate ion is slightly stronger (for example, $\log\beta$ for $\text{Am}(\text{NO}_3)_2^{2+}$ is 0.063 at room temperature, 1 M ionic strength) than that of hexavalent actinyl ions. A simple calculation suggests that >50% of the trivalent actinide and lanthanide ions will be complexed with nitrate ion in 3 M HNO_3 .

It should be noted that Ce(III) would be oxidized into Ce(IV) in our final permeation test, so information on the complexation of Ce(IV) with nitrate ion is also important to understand the permeation behavior of Ce. Since there are no reliable data for the stability constants for Ce(IV)/nitrate complexes, we employed the data for Pu(IV)/nitrate complexes (*Appl. Spectrosc.* 2000, 54, 812-823; *J. Radioanal. Nucl. Chem.*, 1998, 235, 25-29) to conduct the calculation. In general, Pu(IV) would form both 1:1 and 1:2 complexes with nitrate ions under common conditions and the complexation is apparently stronger ($\log\beta_1$ of 0.6 and $\log\beta_2$ of 1.3) than that of trivalent and hexavalent actinides/lanthanides. Calculation results suggest that >90% of the tetravalent Ce ions will be in the form of $\text{Ce}(\text{NO}_3)_2^{2+}$ in 3 M HNO_3 .

As we have mentioned earlier, upon complexation of nitrate ions, the size of metal complexes will change as compared to the hydrated ions. However, the detailed influence imposed by nitrate complexation on the ions' size is not easy to illustrate. The nitrate ion can complex with the metal ions either in monodentate or in bidentate mode, and the number of water molecules replaced in these two situations will be different. Moreover, the complexation with nitrate is expected to cause obvious disturbance to the second hydration shell of the metal ions and thus affect their effective size in the solution. In general, we believe complexation with nitrate ions may increase the size of the metal ions due to the relatively large size of nitrate ions than water molecules. However, to fully address these issues, great efforts must be required at both the experimental and theoretical levels. Currently, we expect the complexation of nitrate ion may impose similar effect on the size of trivalent lanthanides and actinyls, because these metal ions only form 1:1 complex with the nitrate ion under our experimental conditions. But for tetravalent lanthanides or actinides such as Ce(IV) and Pu(IV), the impact of nitrate complexation will be more obvious due to the strong complexation and high complexation stoichiometry (1:2). And this stronger impact has indeed been demonstrated by our experimental results, i.e., the permeation rate of both Ce(IV) and Pu(IV) is apparently slower than that of trivalent lanthanides in natural permeation without the presence of TODGA extraction in the receiving solution (**Figure R6** above and **Figure 3b** in our revised manuscript). (Note: in the presence of TODGA extraction, that will be another story because the extraction of the metal ions by TODGA will weight in and play a significant role in determining the permeation rate/ratio.)

Relevant discussions on nitrate complexation have been provided in supporting information in our revised manuscript.

Comment 7. *Authors incorrectly assign diffusion of ions across GOMs to osmotic effect. Citing the paper: "As shown in Fig. 3b and 3c, the smaller lanthanide ions (8.74 - 9.20 Å) in the FC could pass through the GOM ($\mu = 10.5 \text{ \AA}$) into the RC as a result of the osmosis effect."*

Permeation of ions in the setup used by authors is simple diffusion driven by concentration gradient. Adding extracting liquid helps to reduce concentration of ions on the permeate side and to maintain high gradient. Osmotic effect is related to flow of water from permeate side towards ions not capable to cross the membrane, thus diluting the feed solution. Authors cite in several places “osmosis conditions” of their experiments. But was the osmotic effect actually observed? I wonder if osmotic rise was observed on the feed side when the size of ions was too high for permeation across the membrane, please take pictures as a function of time if that is the case. Was the dilution of not penetrating ion solutions due to osmotic effect registered using any of used concentration measurement methods? If the osmotic rise is not observed, it might be indication of complete chemical blocking of permeation channels by adsorbed ions rather than sieving effect.

Response: We are sorry for not making this point clear in our original manuscript. We have wrongly assigned the diffusion of ions across GOMs to the osmotic effect. We did not observe any osmotic rise in our permeation experiments even using high concentration of uranyl in the feeding side. Meanwhile, we don't think the adsorption of either lanthanide or actinide ions plays an important role in the permeation process because the adsorption of these ions by the GOM is negligible (**Figure R1**) under the strong acidic conditions in our experiments. We have revised all the corresponding descriptions in our revised manuscript.

Reviewer #2:

This manuscript discusses the application of graphene oxide membranes to achieve a lanthanide/actinide separation based on size discrimination between the larger, highly oxidized actinyl ions (AnO_2^n , where n is +1 or +2) versus the smaller Ln^{3+} ions. To accomplish this, a semi-permeable membrane was formed by coating a porous nuclear track membrane with graphene oxide membranes. The separation was driven by osmosis from a concentrated solution to a solution free of ions, the smaller Ln^{3+} ions traversed the membrane more readily than the larger AnO_2^n ions. To enhance the separation, a liquid-liquid extraction system was added to the receiving solution. The manuscript is generally well written, with no obvious grammatical issues. However, there are some issues, which need to be addressed before this manuscript can be recommended for publication. I recommend publishing after minor revisions.

Response: We appreciate greatly the reviewer's positive comment on our work. We have addressed all the issues raised by the reviewer and the details are listed in the following responses.

Comment 1. *This is a very interesting approach and seems to have promise, however the kinetics of the diffusion across the membrane is very slow, taking at least 5 h to reach a 50% removal of the Ln^{3+} and over 24 h to get close 90%. This raises questions about the practicality of the approach.*

Response: Indeed, we acknowledge the slow permeation kinetics is really a problem for our separation approach. Even though we have greatly improved the permeation kinetics of the lanthanides by employing a TODGA extraction system in the receiving side, the diffusion rate is still somewhat slow. We have conducted a literature survey and found that the slow diffusion rate is a common issue for natural permeation of metal ions through GOM (*Sep. Purif. Technol.*, 2017, 188, 523-529; *Nat. nanotechnol.*, 2017, 12, 546-550; *Nature*, 2017, 550, 380-383; *J. Membrane Sci.*, 2017, 521, 1-9). We expect there are several ways to enhance the permeation rate, including to adopt a thinner GOM, to create more defects on the GO sheets, or to impose pressure on the feeding side, etc. However, these measures may act as a double-edged sword, i.e., it could enhance the permeation rate of the lanthanides and meanwhile may also lead to higher diffusion of the actinyl ions. We believe more in-depth investigations are required to find an optimal measure that could balance the diffusion rate and the An/Ln separation factor, and we will definitely continue to work on this issue in the future.

Comment 2. *Following the concerns with long incubation times, how resistant are the membrane to radiolytic degradation, the dose from actual used nuclear fuel solutions are very high?*

Response: Thanks for pointing out this important issue. It is true that the irradiation from actual used nuclear fuel is very strong and any reagents or materials used for the separation may undertake high irradiation dose in real operations. To probe the irradiation stability of the GOM we used in this work, we have irradiated the GOMs with gamma-irradiation and then performed permeation tests with the irradiated GOMs. Specifically, three GOM samples were immersed in 3.0 M HNO₃ separately in glass tubes and then the tubes were irradiated by gamma rays with a dose rate of 120 ± 5 Gy/min at room temperature under air atmosphere using a ⁶⁰Co source (Institute of Nuclear and New Energy Technology, Tsinghua University). The accumulated irradiation doses for the three samples were 10, 50 and 100 kGy, respectively. Then, we performed permeation tests for the sieving of Ce(IV), Nd(III) and U(VI) ions in 3.0 M HNO₃ using the three GOMs. As shown in **Figure R8**, the permeation performance of the three GOMs are essentially the same and are well comparable with the GOM without irradiation (**Figure 3e** in the main manuscript), which indicates that the GOM prepared in our work has excellent gamma-irradiation stability.

Figure R8. Sieving of Ce(IV), Nd(III) and U(VI) in 3.0 M HNO₃ as a function of time through gamma-irradiated GOM of different dose. (a) diagram of sieving setup; (b-d) GOM with irradiation dose of 10 kGy, 50 kGy and 100 kGy, respectively. Initial concentration of Ce, Nd and U is 6 mM, 6 mM and 4 mM, respectively.

Moreover, we also characterized the irradiated GOM samples by XRD, XPS and Raman spectroscopy (**Figure R9**). Obviously, there are no apparent differences between the XRD patterns of the un-irradiated GOM and the gamma-irradiated GOM of different absorbed dose. The inter-layer spacings remain essentially unchanged by irradiating the GOM in 3.0 M HNO₃. The XPS and Raman spectra of the irradiated and un-irradiated GOM samples are also almost identical to each other. Such findings further prove the stability of the GOM against gamma-irradiation.

Relevant results and discussion have been provided in our revised manuscript.

Figure R9. (a) XRD patterns, (b) Raman spectra and (c) XPS spectra of gamma-irradiated (10, 50, 100 kGy) and un-irradiated GOM. The dose rate is 120 ± 5 Gy/min. All the GOMs were irradiated in 3.0 M HNO_3 .

Comment 3. *The separation is not actually that efficient, in the best case there is still 5–10% of the Ln in the starting solution and 5–10% of the An in final solution.*

Response: Yes, as pointed out by the reviewer, although most of actinides and lanthanides are separated from each other, there are still apparent cross contaminations in the An product and Ln product after separation. The lower-than-ideal rejection ratio can be caused by several factors (please also see our responses to the comments of reviewer 1 and to the next comment). To improve the mutual decontamination factors for An and Ln in our work, a more delicate control of the GOM structure is necessary and we are currently working on this point in our lab.

Moreover, we should point out that the current separation efficiency was achieved in a single stage apparatus. Theoretically, the separation efficiency can be significantly improved by employing multiple-stage sieving. For example, the feeding solution and the receiving solution can be further sieved separately by the GOM. With such operations, the amount of An in the Ln product and that of Ln in the An product would further decrease by about ten folds, thus producing Ln and An products with much higher purity. This point has also been raised in the conclusion part in our original manuscript.

Comment 4. *What is the proposed mechanism for the movement of the actinyl ions from the starting solution to the end solution, there seems to be too high of a yield just based on diffusion through the membrane. Also, the waste stream (the receiving solution) has a relatively high actinide concentration, this does not seem conducive for a final disposition of the nuclear waste.*

Response: This is indeed an important question. As we presented in our response to reviewer 1, we attribute the high diffusion yield (5-10%) of the actinyl ions mainly to the following four possibilities: 1) There are pinhole defects on GO sheets and these pinholes (usually with a diameter of several Å to tens of Å) will provide additional pathways for the metal ions to permeate through GOM. 2) The stacking of GO sheets in GOM is usually imperfect and there

are apparent stacking voids in the GOM, which could also allow the actinyl ions to pass through. The stacking voids in GOM and defects on GO have been well recognized in previous reports (*Adv. Mater.*, 2010, 22, 4467-4472; *ACS Nano*, 2018, 12, 7855-7865). 3) The linear actinyl ions may undertake a horizontal configuration relative to the GO sheet to pass through the nanochannels in the GOM. The horizontal configuration is less sterically constrained in the channel and may meet less resistance as compared to the vertical one when passing through the channels. 4) Some of the actinyl ions may undergo dehydration when contact with the GO sheet and their size becomes smaller and thus can pass through the GOM. It is well known the functional groups such as carboxylate on the GO can capture and dehydrate metal ions in aqueous solution (*Phys. Chem. Chem. Phys.*, 2013, 15, 2321-2327; *Chem. Eng. J.*, 2012, 210, 539-546; *Environ. Sci. Technol.*, 2012, 46, 6020-6027; etc). Nevertheless, we believe this dehydration effect should be very limited under the highly acidic conditions in our permeation experiments. As we have presented in our response to the comments of reviewer 1, the adsorption of U(VI) by GO is almost negligible in 3 M HNO₃ due to the protonation of the functional groups on the GO sheets.

Moreover, we should point out that the slightly higher diffusion ratio of Am and Pu is supposed to be caused by the reduction of Am(VI) and Pu(VI) in the sieving process. The unstable hexavalent Am, Pu and Np ions are subject to reduction much more easily than U(VI) and could be reduced to their stable oxidation states (Am(III), Pu(IV) and Np(V)). And the smaller and spheric Am(III) and Pu(IV) ions would pass through the GOM along with the trivalent/tetravalent lanthanides and can be readily extracted by the TODGA ligand in the receiving compartment.

For the issue of high actinide concentration in the waste stream, as we have discussed in our response to the previous comment, we expect that the decontamination efficiency can be further improved by optimizing the structure of the GOM as well as by using multiple-stage sieving technique.

We have substantially enriched our discussion on permeation mechanism of both the lanthanides and actinides in our revised manuscript.

Comment 5. Finally, the authors give no comparison of their results to other methods trying to achieve a similar separation, this is needed. For example see *Ind. Eng. Chem. Res.* 2014, 53, 1624–1631 or *Inorg. Chem.* 2016, 55 (17), 8913-8919.

Response: We appreciate the reviewer's kind reminder. We have consulted relevant references and provided comparison on the separation results between our work and previous reports. In general, there are two types of methods to achieve group separation of An from Ln or other fission products. One is through solvent extraction and the other is through crystallization or precipitation. A brief comparison of relevant work is shown in **Table R1** below.

For the solvent extraction methods, the separation can be achieved either by using a combination of different ligands (such as in *Ind. Eng. Chem. Res.*, 2014, 53, 1624-1631) in tandem extraction cycles or by using a single extractant (such as in *Inorg. Chem.*, 2014, 53, 1712-1720). For example, a series of ligands (TODGA, HEH[EHP], DTPA, CyMe₄-BTBP, etc.) have been used in the GANEX and ALSEP processes. These processes have been relatively well developed and even been demonstrated with centrifugal contactors in several occasions.

High separation factors can be obtained through delicate selection of the ligands and elaborate design of the extraction processes. Nevertheless, the complexity with multiple compounds and many extraction/scrubbing/stripping stages in these processes may cause difficulties in real applications. In recent years, researchers have also designed and synthesized a soft-hard donor combined ligand (Et-Tol-DAPhen) for the group separation of An from Ln (*Inorg. Chem.*, 2014, 53, 1712-1720). Preliminary results indicate that U(VI), Am(III), and Pu(IV) can be separated simultaneously from the lanthanides with this ligand. However, the poor solubility of this ligand in aliphatic diluents and its possible low selectivity to Np could impede its further application in An/Ln group separation.

For the crystallization or precipitation method, the concept is to first oxidize the actinides into the hexavalent actinyl ions and then co-crystallize or co-precipitate these actinyl ions from the lanthanides or other fission products. For example, Burns et al. oxidized Np, Pu, and Am to hexavalent actinyl ions with NaBiO_3 and then co-crystallized them with U(VI) in the form of $\text{AnO}_2(\text{NO}_3)_2 \cdot 6\text{H}_2\text{O}$, while the lanthanides and other fission products will mostly remain in the solution (*Inorg. Chem.*, 2016, 55, 8913-8919). Shehee et al. explored the separation of Am(VI) and other actinides from the lanthanides by oxidizing Am(III) with $\text{Na}_2\text{S}_2\text{O}_8 + \text{AgNO}_3$ and then selectively precipitating the trivalent/tetravalent lanthanides with sulfate ions (*Sep. Sci. Technol.*, 2010, 45, 1743-1752). The above efforts provide a potential and important method for An/Ln group separation. But a major demerit with this method is the poor selectivity, especially for Am. Moreover, this method may encounter criticality risk due to the enrichment of high concentrations of Pu in the crystals or precipitates.

Based on the above comparisons, we could see that our work provides another approach different from the above two methods to achieve An/Ln group separation. Our approach could achieve appreciable An/Ln separation through simple coupling of oxidation, extraction and sieving. The overall An/Ln separation factors in our work are mostly higher than the single stage separation factors of the extraction method and far outperform those of the crystallization or precipitation method. We have added relevant descriptions about these comparisons in our revised manuscript.

Table R1. Comparison of An/Ln group separation in different reports.

Method	Brief descriptions	Metals	Single stage SF (D_A/D_B)	Ref.
Extraction	GANEX process: co-extraction with TODGA/DMDOHEMA and selective stripping with SO ₃ -Ph-BTP	U, Np, Pu, Am, Lns, etc.	$SF_{Ln/Am} = 50 \sim 100$	1, 2
Extraction	ALSEP process: coextraction with TODGA/HEH[EHP] and selective stripping with DTPA	Am, Lns	$SF_{Eu/Am} = 50 \sim 90$ $SF_{Nd/Am} = 39$	3
Extraction	Selective co-extraction with Et-Tol-DAPhen	U, Pu, Am, Eu, Pm, etc.	$SF_{U/Eu} = 277$ $SF_{Am/Eu} = 67$ $SF_{Pu/Pm} = 657$ No Np separation	4, 5
Oxidation + Crystallization	Oxidation with NaBiO ₃ and co-crystallization as AnO ₂ (NO ₃) ₂ ·6H ₂ O	U, Np, Pu, Am, Zr, Nb, Cs, Ce, etc.	$SF_{An/FPs} = 0.99 \sim 7.7$	6, 7
Oxidation + Precipitation	Oxidation with AgNO ₃ + Na ₂ S ₂ O ₈ and precipitation with sulfate	U, Np, Pu, Am, Eu, etc.	$SF_{An/Eu} = 10 \sim 11$	8
Sieving +Oxidation/ Extraction	Oxidation with NaBiO ₃ and then sieving through GOM with the help of TODGA extraction	U, Np, Pu, Am, Ce, Nd, Eu, Gd	$SF_{An/Ln} = 100 \sim 400$	This work

1. Note: There are different versions of the GANEX process and only a latest one was cited here.

2. *Radiochim. Acta*, 2019, 107, 917-929

3. *Ind. Eng. Chem. Res.*, 2014, 53, 1624-1631

4. *Inorg. Chem.*, 2014, 53, 1712-1720

5. *Sep. Purif. Technol.*, 2019, 223, 274-281

6. *Inorg. Chem.*, 2016, 55, 8913-8919

7. *Ind. Eng. Chem. Res.*, 2020, 59, 4756-4761

8. *Sep. Sci. Technol.*, 2010, 45, 1743-1752

Reviewer #3:

In this communication, authors have reported their research pertaining to the selective separation of lanthanides from actinides after oxidizing the later. They have used tailor made GO membrane as a support for the separation of metal ions facilitated by TODGA extractant. After reading the manuscript many times, I did not find any new breakthrough research in this article that qualifies its publication in "Nature Communication". Off course, this is a nice piece of work and may be published as a regular full article in other relevant journal. My decision is based on following points.

Response: We appreciate the reviewer's comment, especially the judgement that "this is a nice piece of work". We understand the reviewer's concerns on the novelty and importance of our work, and we have addressed these concerns by further explicating the highlights of our findings. Details are provided in the following responses to the reviewer's comments. We hope the reviewer now can agree that our revised manuscript fulfills the requirements for publication in *Nat. Commun.*

Comment 1. *Preparation of GO with desired inter-layer spacing is not a new concept. Indeed, it has been prepared earlier also by the authors group and the same has been referred in this manuscript (ref- 18 – 20).*

Response: We appreciate and understand the reviewer's point. We agree that the preparation of GO with desired inter-layer spacing is not a completely new concept and it has been investigated in quite a few research laboratories. However, the tailor-made GO membrane has rarely been applied to the area of nuclear waste treatment, especially for the separation of highly radioactive transuranic elements, the research of which has long been a hot frontier as well as a significant challenge in nuclear chemistry. In this work, we have greatly extended the application scope of GO membrane to dealing with the seminal challenge of group actinide/lanthanide separation, especially under highly acidic conditions. With delicate control of the interlayer spacing of GO, we have achieved unprecedented An/Ln group separation efficiency in a single stage. Indeed, this strategy requires further improvements to enhance its practicability in real applications, but we believe our findings in this work point out a new direction to address the challenges in nuclear waste treatment and we expect with strong confidence that the publication of our work can inspire more experimental and theoretical efforts to deal with this problem.

In addition, we have also further improved the preparation/characterization protocol for the GOM in this work. For example, we have employed for the first time a nuclear-track membrane as the support for the GOM. The nuclear-track membrane provides mechanical support for GOM and meanwhile has abundant large pores/tunnels to allow the quick pass of solutions. And thanks to the suggestions by other reviewers, in our revised manuscript we have performed a systematic characterization of the GOM under different experimental conditions and investigated the effect of metal permeation and radiolytic irradiation on its structure, which has not been well addressed in previous reports. All the results suggest our GOM is very robust in dealing with highly acidic and radioactive solution.

Comment 2. *GO membrane was tested with lanthanides and actinides (in different oxidation state). What is the origin of the mixer of lanthanides and actinides (U, Pu, Np and Am) feed? If authors target is lanthanide/actinide separation, then it should be lanthanides and Am only? If their target is the separation of all lanthanides from a series of actinides, then they should also comment on the selectivity of other fission element and process chemicals, such as Cs, Sr, Fe, Pd etc, which I am sure, will pass through this GO membrane.*

Response: The reviewer is very careful and deep thinking and we appreciate the questions raised. The incentive for An/Ln group separation in this work is inspired by the so-called group

actinide extraction (GANEX) concept, which is supposed to recover all the actinides simultaneously from highly acidic spent fuel solution in a single-step separation, aiming towards the homogeneous recycling of plutonium and minor actinides in a future closed fuel cycle. However, implementation of such a single-step separation concept is not easy due to the great chemical complexity of the actinides and the presence of a large amount of various fission products. In particular, the presence of lanthanides is problematic because these 4f-elements are the major components of the fission products and exhibit great chemical similarity to the trivalent actinide Am, thus impeding the implementation of group separation strategy for all the actinides. For other fission products, the separation is relatively easy considering their bigger differences from the actinides in terms of charge density, ionic size, redox ability, etc. Therefore, we put our emphasis on the An/Ln group separation in this work.

Moreover, based on the experimental results of our work, the current strategy is surely able to recover Am from an Am/lanthanides mixture. On the other hand, if we deal with the original spent fuel solution consisting of actinides, lanthanides, and other fission products, we believe the strategy in our work is capable of separating the actinides from most of the fission products such as Cs, Sr, Fe, Zr, Pd, etc., because these smaller and spherical ions will pass through the GO membrane, as exactly pointed out by the reviewer. To further confirm this, we have conducted permeation tests for these metal ions and the experimental results (Figure R10) do support our judgement. Specifically, In the presence of the TODGA extraction system, the permeation ratio of the metals follows the trend of their affinity to TODGA ($Zr(IV) > Sr(II) > Pd(II) > Fe(III) > Cs(I)$).

Since the current work focus mainly on the An/Ln group separation, we do not include these results in our manuscript. Nevertheless, we have commented on this point in the conclusion part in our original manuscript.

Figure R10. (a) Setup for ion sieving through GOM with the assistance of extraction and (b) the permeation ratios (R) of Zr(IV), Fe(III), Pd(II), Sr(II) and Cs(I) as a function of time. $[HNO_3] = 3$ M, $[TODGA] = 0.1$ M, initial metal concentration in the feeding side: 5 mM of each ion.

Comment 3. Due to slow natural diffusion of metal ions within the pristine GO membrane, authors have used TODGA as the carrier extractant. Therefore, the overall separation becomes a carrier facilitated transport, and the selectivity amongst the metal ions will be decided by the carrier ligand, not by the membrane architecture. If it is so, then proposed methodology has no advantage of using specially designed GO membrane. Instead, it will be a poor choice for the researchers.

Response: It is surely true that the carrier extractant TODGA in the receiving side could facilitate the transport of metal ions (in particular for the trivalent/tetravalent ions) by increasing the concentration gradient of the metal ions across the GO membrane. With the help of the TODGA extraction system, the permeation kinetics of our separation system has been greatly improved, e.g., 93% of Nd(III) will pass through the GO membrane in 24 hours with TODGA extraction while the ratio is only 38% during the same duration in the absence of this ligand. However, we may not be able to agree with the reviewer's judgement "the selectivity amongst the metal ions will be decided by the carrier ligand, not by the membrane architecture". In our separation strategy, we are confident it is mainly the GO membrane providing the An(VI)/Ln(III) selectivity through the sieving effect, i.e., the spheric and smaller Ln(III) can pass through the GO membrane while the linear and larger An(VI) ions will be blocked by the membrane. In fact, even without the employment of the TODGA extraction system in the receiving side, the permeation with the GO membrane can still afford appreciable An/Ln separation (**Figure 3a & b** in the main manuscript, but such a separation is far from satisfactory because of the small permeation ratio and slow diffusion rate of the lanthanides).

On the other hand, we acknowledge TODGA itself does show some selectivity to tetravalent and trivalent lanthanides/actinides over hexavalent actinides according to quite a few previous reports (*J. Nucl. Sci. Technol.*, 2007, 44, 405–409; *Solvent Extr. Ion Exch.*, 2013, 31, 401-415 etc.). We have also conducted additional permeation tests using the support membrane (nuclear track membrane) to prove such a selectivity. As shown in **Figure R11a/b** below, a permeation test with only the support membrane will allow all the selected metal ions (Nd(III), Ce(IV) and U(VI)) pass through the membrane with similar rates. When we applied a TODGA extraction system in the receiving side (**Figure R11c/d**), the permeation of all the three ions accelerated but to a different degree, i.e., Nd(III) and Ce(IV) permeate faster than U(VI). This would result in somewhat separation of U(VI) from the lanthanides. However, the separation between the two groups of ions is very limited because most of U(VI) will pass along with the lanthanides into the receiving side finally. Moreover, as we will present in our responses to the reviewer's following comments, the TODGA-based supported liquid membranes or polymer inclusion membranes are also not able to achieve the desired separation of actinides from lanthanides in group, particularly it cannot separate high valent Am from the lanthanides.

Therefore, the introduction of the TODGA extraction into the membrane sieving system in our work is an innovation and it provides a significant chemical driving force to promote the separation kinetics, but we believe it does not help much with enhancing the An/Ln selectivity.

Figure R11. (a, b) Setup for permeation test using nuclear track membrane and the permeation ratios (R) of Nd(III), Ce(IV) and U(VI) as a function of time; (c, d) Setup for permeation test using nuclear track membrane with the assistance of TODGA extraction and the permeation ratios (R) of Nd(III), Ce(IV) and U(VI) as a function of time. $[HNO_3] = 3$ M, $[TODGA] = 0.1$ M, initial concentrations of Ce, Nd and U in the feeding side are 6 mM, 6 mM and 4 mM, respectively

Comment 4. *The results with TODGA/GO membrane also shows about 8-10% permeation of Am, Np and Pu (dioxocations) along with lanthanides. Therefore, this level of contamination of in the lanthanide product can not be said as selective permeation.*

Response: The reviewer has made an important point. As pointed out by this reviewer as well as by reviewer #1 and #2, although majorities of actinides and lanthanides are separated from each other with our separation strategy, there are still 8-10% permeation of Am, Np and Pu (dioxocations) along with lanthanides. As we have presented in our responses to Comment 3 of reviewer 1 & Comment 4 of reviewer 2, we can generally attribute the permeation of the actinyl ions to the following factors: 1) Pinhole defects (usually with a diameter of several Å to tens of Å) on GO sheets; 2) Imperfect stacking of GO sheets in GOM; 3) Horizontal pass of the linear actinyl ions through the nanochannels in the GOM; 4) Dehydration of the actinyl ions; 5) Reduction of the actinyl ions (for Am and Pu). To improve the single stage decontamination factors for An/Ln in our work, we believe a more precise control of the GOM structure is necessary and we will continue working on this point in our lab. We have provided relevant discussion in the “Mechanistic insights” section in the revised manuscript

Moreover, we should point out that the current separation efficiency was achieved in a single stage apparatus. Theoretically, the separation efficiency can be significantly improved by employing multiple-stage sieving. For example, the feeding solution and the receiving solution can be further sieved separately by GOM. With such operations, the amount of An in the Ln product and that of Ln in the An product will further decrease by about ten folds, thus resulting in Ln and An products with much higher purity.

Comment 5. Separation factor of lanthanides over UO_2^{2+} are in the range of 100 - 400 (maximum). In fact, such a separation factor can also be obtained with TODGA supported polymeric membranes where TODGA itself has poor selectivity for UO_2^{2+} cation over Am and lanthanides (*Dalton Trans.*, 2015, 44, 515-521, *Solv. Extr. Ion Exch.*, 2001, 19, 91-103). This aspect should have been highlighted in the manuscript and reasoning should have been given about the advantages of using such a sophisticated GO membrane over the conventional polymeric membrane.

Response: We appreciate the reviewer's comment and believe the reviewer has raised an important point. Indeed, as pointed out by the reviewer, the ligand TODGA itself has poor selectivity of U(VI) over Am and lanthanides. For example, in a 0.1 M TODGA/*n*-dodecane – 1 M HNO_3 extraction system, the distribution ratio of U(VI) is 0.8, while those of Am(III) and Eu(III) are 30 and 265, respectively (*Solv. Extr. Ion Exch.*, 2001, 19, 91-103). In this regard, the TODGA ligand in principle could be able to separate the trivalent lanthanides from hexavalent actinides in group by solvent extraction under certain conditions. However, this concept encounters great difficulty in real operations due to the reduction of the hexavalent actinides by the organic reagents used for solvent extraction, especially for Am(VI). We once have conducted the extraction experiments for Am(VI) (prepared by $NaBiO_3$ oxidation) from 3.0 M HNO_3 solution by 0.1 M TODGA/*n*-dodecane. As soon as the contact of the two phases, the distribution ratio of Am reaches 112, indicating the fast reduction of Am(VI) to Am(III). The reduction issue has also been confirmed by in-situ spectroscopic monitoring of the two phases during the extraction process. The results in **Figure R12** apparently suggest the fast reduction of Am(VI) to Am(III). As a result of this reduction issue, it is impossible to separate Am from the lanthanides by TODGA extraction.

Figure R12. Absorption spectra of the aqueous and organic phases before and after the extraction of ~ 0.1 mM Am(VI) from 3 M HNO_3 by 0.1 M TODGA/*n*-dodecane. Contact time: 10 s.

On the other hand, thanks to the reviewer's reminder, TODGA and related ligands can be employed as carrier extractants in supported liquid membrane or polymer inclusion membrane, and these membranes have been used for the recovery of a variety of metal ions such as Am(III), Pu(IV), Ln(III), Sr(II), etc. (*Ind. Eng. Chem. Res.* 2009, 48, 8605-8612; *J. Hazard. Mater.*, 2011, 188, 281-287; *J. Hazard. Mater.*, 2012, 237-238, 339-346; *J. Hazard. Mater.*, 2014, 275, 146-153.). Considering TODGA's intrinsic selectivity to Ln(III) over An(VI), it is possible to use the membrane technique to separate Ln from An. To demonstrate this possibility, we have prepared four typical TODGA-included membrane ([1] Polytetrafluoroethylene (PTFE, 0.2 μm) membrane; [2] Polytetrafluoroethylene (PTFE, 0.45 μm) membrane; [3] Polypropylene membrane; [4] Membrane with cellulose triacetate (CTA) as the polymer matrix and 2-nitrophenyloctyl ether (NPOE) as the plasticizer) according to established procedures in previously published reports (*Same references as above*) and studied their performance in An/Ln separation. To facilitate the analysis, here we only selected Am/U and Eu as the representatives for An and Ln, respectively.

First, we performed transport tests by a starting feed solution pre-oxidized with NaBiO_3 (the NaBiO_3 solid was filtrated out before the transport test). As shown in **Figure R13**, with all the four membranes, the transport rate of U is slower than that of Am and Eu, thus achieving certain degrees of separation of U from Am and Eu. These results are well consistent with the literature. However, all these systems are not able to achieve effective separation because the significant transport of U from the feeding side to the receiving side. More importantly, although we pre-oxidized Am(III) to Am(VI) before the transport tests, the transport behavior of Am well resembled that of Eu, indicating the oxidized Am(VI) ions have been reduced to Am(III) quickly when contacted with the membrane. Therefore, no separation of Am from the lanthanides can be achieved through these transport approaches.

Figure R13. Transport tests by membranes containing TODGA as the carrier extractant. 3 M HNO_3 in the feeding solution and 0.01 M HNO_3 in the receiving (stripping) solution. The feeding solution was pre-oxidized by NaBiO_3 and then the solid was filtrated out.

To overcome the reduction issue of Am(VI), we then conducted transport tests by adding solid NaBiO₃ in the feeding solution, which is similar to the operation in our ion sieving system. As shown in **Figure R14**, the transport behaviors of Eu and U are essentially the same as that in **Figure R13**, while the transport rate of Am was slowed to certain degrees. These results suggest Am(VI) has been somewhat stabilized with the presence of NaBiO₃ in the feeding solution. Nevertheless, the presence of NaBiO₃ is still unable to sufficiently stabilize Am(VI) because the transport behavior of Am obviously resembled more closely with that of Eu(III) than U(VI). We attribute the insufficient stabilization of Am(VI) to the direct contact of the high valent Am with the organic compounds (the polymers, TODGA, etc.) in the membrane.

Figure R14. Transport tests by membranes containing TODGA as the carrier extractant. 3 M HNO₃ in the feeding solution and 0.01 M HNO₃ in the receiving (stripping) solution. Solid NaBiO₃ was constantly present in the feeding solution.

Based on the above transport experiments, we could see the TODGA-included membranes do show a certain degree of Ln/An selectivity but these membranes are unable to achieve the group separation of An and Ln, especially for the separation of Am from lanthanides. In contrast, our GOM-based ion sieving system can achieve appreciable An/Ln group separation efficiency by taking advantage of the ion selection effect of GOM, and the aid of TODGA extraction that help further speed up the kinetics. Moreover, the separation strategy in our work offers a perfect mechanism to overcome the reduction issue of high valent actinides (especially for Am) because the actinides in the feeding solution have no chance to directly contact with the organic reagents in the receiving solution. Moreover, the GOM itself can hardly reduce Am(VI) in the presence of NaBiO₃ during the permeation test (please see details in our response to comment 2 of reviewer 4).

To address the reviewer's concern, we have added additional discussions on this issue and cited a few relevant literatures (*Ind. Eng. Chem. Res.* 2009, 48, 8605-8612; *J. Hazard. Mater.*, 2012, 237-238, 339-346; *J. Hazard. Mater.*, 2014, 275, 146-153.) in our revised manuscript.

Comment 6. *The 90% transport of lanthanides took about 24 h in TODGA assisted GO membrane in this work. This can be achieved in less than 4 hrs if one uses a polymeric membrane support for TODGA. Authors have not highlighted this fact in their manuscript.*

Response: First, we acknowledge the slow permeation kinetics with GO membrane is currently a headache issue for application of the separation approach in our work. Even though we have greatly improved the permeation kinetics of lanthanides by employing a TODGA extraction system in the receiving side, the diffusion rate is still somewhat slow. We have conducted a literature survey and found that the slow diffusion rate is a common issue for permeation separation through GOM (*Sep. Purif. Technol.*, 2017, 188, 523-529; *Nat. nanotechnol.*, 2017, 12, 546-550; *Nature*, 2017, 550, 380-383; *J. Membrane Sci.*, 2017, 521, 1-9). We expect there are several ways to enhance the permeation rate, including to adopt a thinner GOM, to create more defects in GO, or to add pressure on the feeding side, etc. However, these measures may act as a double-edged sword, i.e., it could enhance the permeation rate of the lanthanides and may meanwhile lead to less rejection of the actinyl ions. To find an optimal measure that could balance the diffusion rate and the An/Ln separation factor, we believe a systematic investigation is required and we will definitely continue to work on this issue in the future.

As pointed out by the reviewer, the polymeric membrane for TODGA does provide a much faster transport rate, which has been reported in the literature (*J. Hazard. Mater.*, 2014, 275, 146-153.) and has also been proved by the transport experiments we conducted in our above response. This fast transport can be attributed to the direct contact of the metals with the TODGA ligands in the membrane and the more porous structure of the polymeric membrane. However, as we have demonstrated above, these polymeric membranes are not able to afford high An/Ln separation factors and to overcome the reduction issue of high valent actinides especially for Am.

Comment 7. *Overall, my impression is that this work does not has sufficient novelty and breakthrough research which can be published in Nature communication. However, undoubtedly, this is a nice piece of work, and can be published as a regular full article in other journal which authors find most suitable.*

Response: We appreciate greatly both the positive and critical comments from the reviewer, and we hope that the above responses have properly addressed the reviewer's concerns and the reviewer now could agree that our revised manuscript fulfills the requirements for publication in *Nat. Commun.*

Reviewer #4:

In this report the authors report using a graphene oxide layer to effectively sieve actinide dioxocations from lanthanide spherical cations. The authors control graphene oxide swelling to an optimal channel size $\approx 10.5 \text{ \AA}$ while adding a chemical driver, solvent extraction, to allow the

smaller lanthanide spherical cations to pass through while the larger actinyl cations are held back in the original feed solution. Graphene oxides have been used in the past to purify water and used to remove heavy metals from solution. This work represents a very clever and relatively easy approach to achieving one of the most challenging separations known to man. It is unclear to this reviewer at this time if this process can be scaled up to process tons of used fuel, but at the very least it can be employed as an analytical technique for preconcentration of lanthanides from mixed matrices. The authors present a well written manuscript with conclusions that are consistent with the data presented. This reviewer recommends publishing but has a few suggestions/questions for the authors to address before publishing.

Response: We appreciate greatly the reviewer's positive comment on our work, especially for recognizing this work "represents a very clever and relatively easy approach to achieving one of the most challenging separations known to man". Also, we fully agree with the reviewer that currently there are still questions about the scale-up of this strategy to process tons of used fuel but it can be employed as an analytical technique for preconcentration of lanthanides from mixed matrices. We have further improved our manuscript by taking consideration of the comments from all the reviewers and hope the revised version could be accepted for publication.

Comment 1. *In the supplemental information the authors show a picture of the experimental apparatus in use with the highly insoluble sodium bismuthate in the FC solution. Did the insoluble sodium bismuthate collect on the GOM surface (on the FC side) and clog the channels that allow the lanthanides pass through? If not, do the authors anticipate this if the experiments are run longer than 24 hours? Do the authors anticipate this to be problematic if the process is scaled up to accommodate more concentrated actinide solutions and thus more sodium bismuthate?*

Response: We appreciate greatly the questions raised. To maintain a strong oxidizing environment in the FC solution, we have added solid sodium bismuthate in the solution. We expect that sodium bismuthate may not collect on the GOM surface and clog the channels based on the following two reasons: 1) The GOM in our apparatus is fixed vertically but not horizontally between two constantly stirred solutions, this would significantly lower the possibility of solid sedimentation on the GOM surface; 2) the permeation rates of the representative lanthanides (Eu, Nd, and U) through the GOM are almost unaffected by the addition of sodium bismuthate, as evidenced by the results in **Figure R15** (data are extracted from the results in **Figure 3** in the main manuscript).

Figure R15. Comparison of ion sieving of Nd^{III} , Eu^{III} , Gd^{III} and U^{VI} through GOM (a) with and (b) without the presence of NaBiO_3 in the feeding solution. (Data are extracted from Figure 3 in the main manuscript).

Nevertheless, to make this point clearer, we have conducted additional experiments to reveal the possible sedimentation of solid sodium bismuthate on GOM. We collected a GOM after 24 h of mixing with NaBiO_3 in a permeation test and gently washed the surface with a small amount of deionized water for one time and then dried it at room temperature. Then we performed SEM and EDS analysis of the GOM surface. As shown in **Figure R16** below, the content of Na and Bi on the surface is very low and close to the detection limit, indicating that the sedimentation of solid NaBiO_3 on GOM is negligible and it won't clog the channels of the membrane.

Figure R16. SEM-EDS analysis results of GOM surface after permeation test in the presence of solid NaBiO_3 .

Comment 2. *The authors state that based on figure 4, with mixed waste, the permeation of the lanthanides reaches >90% and the permeation ratios of actinides are retained at <9%. This implies all actinides behaved the same. They define the separation factor and present the highest separation factor of 400 for U/Ce. Although this conclusion is drawn directly from the results, it is a little misleading since the most important separation factors gained from this study should involve the actinides that require oxidation state control under these conditions, i.e., Np, Am and Pu. Under these conditions, hexavalent uranium will be the most stable actinyl ion. From the 3D bar graph it is hard to determine the differences between R(%) for the actinides. This reviewer suggest the authors provide a more detailed description of the R(%) for each actinide (e.g. alpha counts or concentrations from the LSC counting for the FC and RC solutions for all actinides). Provide a hypothesis to explain why Am, Np and Pu concentrations are increasing in the RC solutions to account for the lower separation factors for all lanthanides compared to U. Are Np, Am, and Pu being reduced back to spherical cations during the 24h permeation period? If so, how can that happen in the presence of excess sodium bismuthate? Are Np, Am and Pu being reduced by the graphene oxide layer upon contact?*

Response: This is surely an important point. Indeed, U(VI) is the most stable actinyl ion while the hexavalent Np, Pu and Am will be subject to reduction more easily and thus require oxidation control to ensure their appreciable separation from the lanthanides. Considering the importance of these transuranic elements in the overall separation, we have provided the detailed R(%) values for each actinide in the updated Fig. 4 in our revised manuscript according to the reviewer's suggestion. Moreover, the permeation trend of the four actinides has been further rationalized in our revised manuscript.

As evidently can be seen from the revised figure, the R(%) for the actinides follows the order of $U \sim Np < Pu < Am$. Such a permeation order is closely related to the redox properties of the four actinides: 1) Am(VI) is the least stable one among the four actinyl ions and it can be easily reduced to Am(III) in solutions. As shown in **Figure R17a** below, with the addition of GO sheets into an Am(VI) solution, both the reducing products Am(V) and Am(III) can be detected in the solution. Approximately 15% of Am will be reduced to Am(III) in three hours. The reduction can be attributed to both the reduction of Am through self-radiolysis and the reduction by GO. 2) Pu(VI) is also not very stable in aqueous solution and it can be reduced to Pu(IV), which can pass through the GOM easily along with the lanthanides. 3) Np(VI) is an actinyl ion worth of more discussions here. In the first place, Np(VI) is even less stable than Pu(VI) (*Chem. Eur. J.*, 2013, 19, 16690-16698). However, the reducing product of Np(VI) is mainly Np(V), which is the most stable form of Np and it is still a linear actinyl ion that can be well blocked by the GOM (see **Figure 3** in the main manuscript). Therefore, the permeation of Am and Pu into the receiving solution is higher than that of U while the permeation of Np is well comparable to that of U.

Figure R17. The evolving of Am absorption spectra as a function of time. (a) In the presence of GO flakes (1.5 mL of 0.03 mM Am in 3 M HNO₃ solution, Am was pre-oxidized by NaBiO₃ and the solid was filtrated out and then 1 mg GO flakes was added). (b) In the presence of GO flakes and NaBiO₃ (1.5 mL of 0.03 mM Am in 3 M HNO₃ solution, 22.5 mg NaBiO₃ and 1 mg GO flakes was added).

Nevertheless, the permeation ratios of Am and Pu (9.0% and 8.4% at 24 h) are only slightly higher as compared to U (7.0% at 24h), suggesting the high valent Am and Pu ions are quite stable in the feeding solution during the permeation process with the help of constant NaBiO₃ oxidation (**Figure R17** above). As shown in **Figure R17b**, in the presence of both NaBiO₃ and GO in the solution, Am(VI) is quite stable and is only slightly reduced over a long term (~3% was converted to Am(III) over a time duration of 24 h). Such an observation is consistent with the slight increase of Am permeation ratio in the permeation test. Therefore, we hypothesize that most of the permeation for all the actinides is caused by the intrinsic defects like pinholes and stacking voids in the GOM (see our details responses to Comment 3 of reviewer 1 & Comment 4 of reviewer 2).

Comment 3. It appears from the 3D bar graph that Eu(III) and Nd(III) are the least permeated lanthanides during the mixed waste test. This could help account for the lowest reported separation factor $Nd/Am = 126$. Can the authors explain the reduced permeation for Eu and Nd in the mixed waste test?

Response: We appreciate greatly the reviewer’s comprehensive thought. We believe the reduced permeation for Eu and Nd as compared to Gd and Ce can be attributed mainly to the different extraction ability of TODGA towards the lanthanides. The extraction of the lanthanide series by TODGA has been well investigated in previous reports (*Hydrometallurgy*, 2020, 195, 105367; *Sep. Purif. Technol.*, 2014, 128, 18-24; *Sep. Sci. Technol.*, 2017, 52, 1006-1014; *Solv. Extr. Ion Exch.*, 2015, 33, 625-641; *Inorganica Chim. Acta*, 2020, 513, 119928; etc.) and it follows the following two trends: 1) heavier lanthanides are more easily extracted by TODGA than the lighter ones (**Figure R18**, copied from *Solv. Extr. Ion Exch.*, 2015, 33, 625-641); ii) tetravalent Ln/An is usually more extractable than the trivalent ones (*Solvent Extr. Ion Exch.*, 2007, 25, 205-224). Consequently, among the four lanthanides studied in our work, Ce(IV) is the most extractable ion, followed by Gd(III), Eu(III) and Nd(III) in sequence. Since the permeation rate (kinetics) of the lanthanides depends significantly on the extraction by TODGA,

Editorial Note: Figure from Sasaki, et al., "The Effect of Alkyl Substituents on Actinide and Lanthanide Extraction by Diglycolamide Compounds", *Solvent Extraction and Ion Exchange*, 33:7, 625-641, DOI: 10.1080/07366299.2015.1087209. Reprinted by permission of Taylor & Francis Ltd

it will thus follow the order of Ce(IV) > Gd(III) > Eu(III) > Nd(III) (**Figure 3** in the main manuscript), with Eu(III) and Nd(III) being the least permeated lanthanides during the mixed waste test. To make this point clearer, we have added relevant discussions in our revised manuscript.

Figure R18. Extraction of lanthanides by TODGA. (Copied from *Solv. Extr. Ion Exch.*, 2015, 33, 625-641)

Comment 4. How robust against alpha radiation are the graphene oxide layers? Can more concentrated actinide solutions with increased alpha decay events destroy the channels or alter the channel size in the graphene oxide layer to reduce efficiency of the separation?

Response: The reviewer has pointed out an important issue, which has also been raised by reviewer 2. It is true that the graphene oxide layers used for the separation would undertake alpha radiation in real operations. Therefore, investigation on alpha radiation effect on the structure of GOM and its separation ability is important to evaluate the practicability of this separation strategy. We believe the best way to do this is conducting the separation experiments using high concentrations (~ several mM) of Am-241 and Pu-239, which contribute most of the alpha radiation in the spent fuel. Unfortunately, we are not able to perform such an experiment currently in our lab due to the extremely radiotoxicity of Am-241 and Pu-239 and the operational limit in our lab.

Nevertheless, to probe the irradiation stability of the GOM we used in this work, we have irradiated the GOM in 3.0 M with gamma-irradiation and then performed permeation test with the irradiated GOM. The details have been presented in our response to comment 2 of reviewer 2. To facilitate the reviewing, we copied here the main conclusions from the irradiation experiments: the permeation performance of the three irradiated GOMs (irradiation dose of 10, 50 and 100 kGy, respectively) are essentially the same and are well comparable with the GOM without irradiation, indicating the GOM prepared in our work exhibits excellent gamma-

irradiation stability. Moreover, XRD and Raman characterization of the unirradiated and irradiated GOM suggests there are no apparent changes in the structure, further proving the robustness of the GOM under gamma-irradiation.

Relevant results and discussion have been provided in the section “Robustness of GOM” in our revised manuscript.

The reviewer feels the manuscript can be greatly strengthened by addressing these suggestions/questions.

Response: We very much thank again for the reviewer’s constructive comments. We have carefully considered all the suggestions and questions and have revised the manuscript accordingly. We hope the revised manuscript could be acceptable for publication.

REVIEWER COMMENTS

Reviewer #1 (Remarks to the Author):

The authors provided detailed reply to my initial comments and revised the text of paper substantially. The revised version of paper has some major issues corrected in my point of view. I still have some disagreements with authors over the suggested mechanism of permeation related to oxidized/not oxidized regions but it should not prevent this paper from publication. Authors made an effort to reflect on alternative point of view and that is sufficient for readers to make own conclusion. Once again I think that experimental part is interesting and important, more detailed elaboration of mechanism of permeation in GO membranes will be clearly settled in next few years with more evidence presented. Therefore, I recommend the paper for publication but provide several discussion points which might be useful for authors.

1. "First, XPS analysis (Figure 2e in our revised manuscript) suggests that the C/O ratio of GO in our work is ~ 2.45 . This value is quite large as compared to common GOs in literature (J. Phys. Chem. C, 2017, 121, 20489-20497; Mat. Today, 2021, 50, 44-54) and indicates the oxidation degree of our GO is relatively small."

The structural model by A.Lerf is the most commonly accepted for GO. This model was designed for GO with C/O in the range 2.4-2.6. That is the oxidation level of "classical" graphene oxide. The Lerf model also did not include large not oxidized regions. Smaller C/O ratio of ~ 2 is very often related to sulphur impurity nearly always present in Hummers GO, as much as 20% of all oxygen in GO can be in sulphate groups. It can also be around 2 in J.Tour method GO but also sometimes related to oxygen from impurities. The GO with C/O below 2.4 was considered in classic studies (A.Lerf et al, I.Dekany et al, earlier papers by Hoffmann or Boehm) as "over-oxidized." I can recommend review chapter by A.Lerf in a book by Z.Eigler and A.Dimiev for more details (Graphene Oxide: Fundamentals and Applications).

The C/O = 2.45 is sufficient to provide 2 oxygen groups per each graphene hexagon. It is hard to imagine that clustering of oxygen groups could provide sufficiently large unoxidized regions to form sizable interconnected network, not to say capillary so that these unoxidized regions overlap on both sides of interlayer. The best paper to provide modelling of oxidized/non oxidized proportion of GO surface vs C/O is to my knowledge that one: DOI: 10.1021/acsami.5b08824 Interconnected graphene regions are found in this study for C/O around 3.3

I also carefully checked Mat.Today paper by Joshi group. Unfortunately the C/O ratio reported in this paper should not be trusted. Using survey spectra for C/O is not commonly accepted procedure, the data shown in the paper also not in agreement with their numbers. The C/O numbers are in direct conflict with shown XPS spectra and clearly not corrected to sulphate oxygen. Most likely the C/O of their (very standard) GO is also around 2.4-2.5 looking at relative intensity of C1s peaks in XPS (C-C and C-O), it is nearly identical to the relative intensity of peaks in the reviewed manuscript.

The paper by Joshi group actually confirms that unoxidized regions are not helping to increase permeation. Prolonged annealing at 80C clearly resulted in partial reduction of their GO, evidenced by their own XRD and XPS spectra (in contrast to their statement), see the change in relative intensity of C-C and C-O peaks in Figure S5. Most important, more not oxidized regions were detected in annealed samples but permeation became slower, not faster. More oxidation gives better swelling and better permeation, not other way around.

Multilayered GO is clearly hydrophilic material. It is impossible to explain its major properties by small fraction of hydrophobic graphitic regions on individual GO flakes. The whole idea of "superfast" flow in graphene capillaries was only a result of idealized and unrealistic for true GO geometrical models.

2. About not perfect sieving. It is well known that interlayer distance in GO is not changing by exact steps related to insertion of water monolayers. The small increments in the interlayer distance observed experimentally were traditionally (starting already from 60-s, then in A.Lerf papers etc) explained by interstratification. In a first approximation, real GO membrane has some interlayers filled with 2 layers of water and some interlayers with 3 layers of water. The true proportion in the relative number of these layers is difficult to estimate, possibly smaller fraction

has 3 layers. What we read from XRD data is AVERAGED value. Note that only one XRD peak is observed in case of random packing of differently hydrated layers. Unfortunately, once commonly accepted in 1960-s and 90-s, interstratification remains to be little known effect among GO membrane researchers, see recent paper on that subject: DOI: 10.1038/s41565-021-01066-0 Therefore, ions which can be perfectly cut off by 2-water layer interlayers can still penetrate 3-water layer filled interlayers. This would explain why ~5% of ions could still permeate the membranes and sieving is not perfect. As a model, random interstratification possibly explains experimental observations.

Reviewer #2 (Remarks to the Author):

"Ion Sieving in Graphene Oxide Membrane Enables Efficient Actinides/Lanthanides Separation"
Zhipeng Wang, Liqin Huang, Xue Dong, Tong Wu, Qi Qing, Yuexiang Lu, Jing Chen, and Chao Xu,
Nat. Commun., NCOMMS-22-30550A

This manuscript has been revised according to feedback given by four reviewers and has been improved. However, a few items still need to be addressed. I recommend this article undergo revisions before it is considered for publication.

Comments:

- 1) Page 6-7 Line 185-196: A SF is normally calculated by taking the ratio the distribution ratios (D-values) of the two species you are trying to separate. The D-value is simply the concentration of the species of interest in the product phase over that in the initial phase. In the current work, the permeation ratios (R), would be equivalent to the D-value. So, by taking the ratio of these, the maximum SF is ~10, which is obvious, as the separation is not actually that efficient, in the best case there is still 5–10% of the Ln in the starting solution and 5–10% of the An in final solution. The authors calculate what they call a "separation factor" (SF), but is actually not an SF. It is something else that is artificially higher due to the numerators of (1-R) in each term. This makes their comparison to other works meaningless, and in fact their value should be one of the low performers.
- 2) Page 7 Line 209-214: The discussion on the stable oxidation states of the actinides and their relationship to the permeability of the GOM is very weak. To begin with, in 3 M HNO₃ the preferred oxidation state of Np is a mixture of Np(V) and Np(VI), with the majority being Np(VI). Second, in the presence of NaBiO₃ both Np and Pu will be completely oxidized to the An(VI) state, along with Am (albeit, Am is a bit more difficult to oxidize). Finally, and most importantly, there was no evidence in the UV-Vis-NIR absorbance spectrum that anything but the An(VI) existed in the presence of NaBiO₃. In light of this, it seems a different reason is necessary to account for the An(VI) species crossing the GOM.
- 3) Page 9 Line 269-279: The proposed mechanism given by the authors is that the ions diffuse through defect in the GOMs occurring where two GO nanosheets come together, which is depicted in Fig. 5. Yet, throughout the manuscript the size of the pores in the nanosheets is what is attributed to the separation. What is the actual root of the separation?

Reviewer #3 (Remarks to the Author):

Comments

Though the authors have given their response very explicitly with the support of few additional experiments, they could not bring out the novelty or any breakthrough finding in this article which can qualify its publication in Nature communication. Even after reading the revise version carefully, I have following concerns over its novelty claim.

1. As claimed in the abstract "Herein, we report a novel and efficient separation", neither

NaBiO₃ is a novel reagent for oxidation of Am(III) to Am(VI), nor the synthesis of GO with desired inter-layer spacing is new. Authors themselves have agreed that these materials are well studied. Then there is no overall novelty which fits for its publication in Nature Comm.

2. Authors claim for applicability of GO/TODGA assisted membrane for treating genuine nuclear waste (page 6 last para) is indeed a wrong statement. In fact, they are only targeting what is transported in the RC side. But, the actual product will remain in the FC side. This feed containing Am(VI) will always be contaminated with lanthanides as 100% transport of lanthanides is not possible.

3. Looking at 100 to 200 times higher concentration of lanthanides as compared to Am in the nuclear waste, even 1% left over lanthanides in the feed will give their concentration more than Am. So this approach will not be good for separation of lanthanides from actinides. Please note: Separation of Pu and U from lanthanides is not important as they can be achieved very easily. But the main concern is separation of Am(III) from lanthanides.

4. Authors have not taken into account of Mo, Tc, Ru in the feed as the concentrations of these metal ions in nuclear waste are orders of magnitude higher than Am. Obviously, being oxycations/oxyanions, these metal ions will not be transported, and will remain with the Am product in FC side. In such a scenario, the product Am will be having more amounts of these metals and non-transported lanthanides than Am itself. Under these conditions how one can claim "an efficient separation method"?

5. Since GO membrane acts as a membrane filter, membrane fouling has not been looked into. In fact, it will never work under nuclear waste condition and membrane pores will be choked in no time. This is an important aspect that must be looked into before proposing any membrane separation process. This aspect is certainly missing in this manuscript.

6. Authors might have tried this membrane for nano or ultra-nano filtration by applying pressure. In such a case, the permeate will be lanthanides and all other non-retained species, whereas product Am(VI) and other elements of similar size will be collected in the reject stream (as depicted in Fig. 1a). In this way, one may expected very fast separation and its practical utility.

Considering above week points, I do not find this manuscript suitable for Nature Communication. However, as I mentioned in my previous recommendation, undoubtedly this is a nice piece of work, and can be published as a regular full article (not as a communication).

Reviewer #4 (Remarks to the Author):

The authors have addressed all my concerns from the previous review in the responses to the reviewers document and revised manuscript. The authors have responded in a very organized and thoughtful manner. I suggest publishing the revised manuscript.

Point-to-point response to reviewers' comments

Reviewer #1

The authors provided detailed reply to my initial comments and revised the text of paper substantially. The revised version of paper has some major issues corrected in my point of view. I still have some disagreements with authors over the suggested mechanism of permeation related to oxidized/not oxidized regions but it should not prevent this paper from publication. Authors made an effort to reflect on alternative point of view and that is sufficient for readers to make own conclusion. Once again I think that experimental part is interesting and important, more detailed elaboration of mechanism of permeation in GO membranes will be clearly settled in next few years with more evidence presented. Therefore, I recommend the paper for publication but provide several discussion points which might be useful for authors.

Response: We are very grateful for the reviewer's insightful comments in two rounds of reviewing, which help greatly enriched our understanding of the permeation behavior of Ln/An in GOM and thus improved our manuscript substantially. The additional comments provided by the reviewer below are also very helpful and we have incorporated some of the points in our newly revised manuscript.

1. "First, XPS analysis (Figure 2e in our revised manuscript) suggests that the C/O ratio of GO in our work is ~2.45. This value is quite large as compared to common GOs in literature (J.Phys. Chem. C, 2017, 121, 20489-20497; Mat. Today, 2021, 50, 44-54) and indicates the oxidation degree of our GO is relatively small."

The structural model by A.Lerf is the most commonly accepted for GO. This model was designed for GO with C/O in the range 2.4-2.6. That is the oxidation level of "classical" graphene oxide. The Lerf model also did not included large not oxidized regions. Smaller C/O ratio of ~2 is very often related to sulphur impurity nearly always present in Hummers GO, as much as 20% of all oxygen in GO can be in sulphate groups. It can also be around 2 in J.Tour method GO but also sometimes related to oxygen from impurities. The GO with C/O below 2.4 was considered in classic studies (A.Lerf et al, I.Dekany et al, earlier papers by Hoffmann or Boehm) as "over-oxidized." I can recommend review chapter by A.Lerf in a book by Z.Eigler and A.Dimiev for more details (Graphene Oxide: Fundamentals and Applications).

The C/O =2.45 is sufficient to provide 2 oxygen groups per each graphene hexagon. It is hard to imagine that clustering of oxygen groups could provide sufficiently large unoxidized regions to form sizable interconnected network, not to say capillary so that these unoxidized regions overlap on both sized of interlayer. The best paper to provide modelling of oxidized/non oxidized proportion of GO surface vs C/O is to my knowledge that one: DOI: 10.1021/acsami.5b08824 Interconnected graphene regions are found in this study for C/O around 3.3.

I also carefully checked Mat.Today paper by Joshi group. Unfortunately the C/O ratio reported in this paper should not be trusted. Using survey spectra for C/O is not commonly accepted procedure, the data shown in the paper also not in agreement with their numbers. The C/O

numbers are in direct conflict with shown XPS spectra and clearly not corrected to sulphate oxygen. Most likely the C/O of their (very standard) GO is also around 2.4-2.5 looking at relative intensity of C1s peaks in XPS (C-C and C-O), it is nearly identical to the relative intensity of peaks in the reviewed manuscript.

The paper by Joshi group actually confirms that unoxidized regions are not helping to increase permeation. Prolonged annealing at 80°C clearly resulted in partial reduction of their GO, evidenced by their own XRD and XPS spectra (in contrast to their statement), see the change in relative intensity of C-C and C-O peaks in Figure S5. Most important, more not oxidized regions were detected in annealed samples but permeation become slower, not faster. More oxidation gives better swelling and better permeation, not other way around.

Multilayered GO is clearly hydrophilic material. It is impossible to explain its major properties by small fraction of hydrophobic graphitic regions on individual GO flakes. The whole idea of “superfast” flow in graphene capillaries was only a result of idealized and unrealistic for true GO geometrical models.

Response: We really appreciate the reviewer’s meticulous analysis of the data in our manuscript and the papers we mentioned. The information not only helps us to further think about the more proper permeation model in GOM, but also inspires us to investigate relevant issues in our future studies. We have read carefully the literatures recommended by the reviewer. We agree with the reviewer that C/O ratio calculated from XPS might not be that accurate and may include the contribution from impurities such as sulphate, and therefore the direct comparison of C/O ratios in different work might be unreliable. For the GO in our work, FT-IR spectra also suggest there are likely traces of sulfate present in the GO sample. Nevertheless, one thing we are sure is the C/O ratio should be relatively high for the GO we prepared in the present work because we have deliberately selected a synthesis protocol to give high C/O ratio based on the synthesis results of our previous work (*Adv. Sci.*, 2021, 8, 2002717). Accordingly, we have further revised our discussions on C/O ratio comparison in our manuscript.

2. About not perfect sieving. It is well known that interlayer distance in GO is not changing by exact steps related to insertion of water monolayers. The small increments in the interlayer distance observed experimentally were traditionally (starting already from 60-s, then in A.Lerf papers etc) explained by interstratification. In a first approximation, real GO membrane has some interlayers filled with 2 layers of water and some interlayers with 3 layers of water. The true proportion in the relative number of these layers is difficult to estimate, possibly smaller fraction has 3 layers. What we read from XRD data is AVERAGED value. Note that only one XRD peak is observed in case of random packing of differently hydrated layers. Unfortunately, once commonly accepted in 1960-s and 90-s, interstratification remains to be little known effect among GO membrane researchers, see recent paper on that subject: DOI: 10.1038/s41565-021-01066-0

Therefore, ions which can be perfectly cut off by 2-water layer interlayers can still penetrate 3-water layer filled interlayers. This would explain why ~5% of ions could still permeate the membranes and sieving is not perfect. As a model, random interstratification possibly explains experimental observations.

Response: We are very grateful for the reviewer's suggestion. We fully agree with the reviewer's opinion that XRD characterization of GOM only shows the averaged layer spacing and the membrane contains uneven water distribution. This uneven water distribution has been verified both experimentally (*Nano Lett.*, 2014, 14, 3993-3998) and theoretically (*ACS Nano*, 2019, 13, 2995-3004). Also, as pointed out by the reviewer, random interstratification might be an important factor to account for the non-ideal rejection of actinyl ions in multilayered GOM, i.e., the interlayers with more layers of water would provide pathways for the larger actinyl ions to pass through. We have provided more discussions on this issue in the section "Mechanistic insights" in our newly revised manuscript.

Reviewer #2

This manuscript has been revised according to feedback given by four reviewers and has been improved. However, a few items still need to be addressed. I recommend this article undergo revisions before it is considered for publication.

1) Page 6-7 Line 185-196: A SF is normally calculated by taking the ratio the distribution ratios (*D*-values) of the two species you are trying to separate. The *D*-value is simply the concentration of the species of interest in the product phase over that in the initial phase. In the current work, the permeation ratios (*R*), would be equivalent to the *D*-value. So, by taking the ratio of these, the maximum SF is ~10, which is obvious, as the separation is not actually that efficient, in the best case there is still 5–10% of the *L_n* in the starting solution and 5–10% of the *A_n* in final solution. The authors calculate what they call a "separation factor" (SF), but is actually not an SF. It is something else that is artificially higher due to the numerators of (1-*R*) in each term. This makes their comparison to other works meaningless, and in fact their value should be one of the low performers.

Response: Thanks for the reviewer's comment. We have carefully checked our calculation method of the separation factor (*SF*) and we are confident that the calculation is correct. As exactly pointed out by the reviewer, a *SF* is normally calculated by taking the ratio the distribution ratios (*D*-values) of the two species you are trying to separate and the *D*-value is simply the concentration of the species of interest in the product phase over that in the initial phase. In this work, as we have defined in our original manuscript, the permeation ratio (*R*) is the percent portion of ions permeated from FC to RC. It is the percentage of ions in the initial FC solution permeated into RC at a given time (For example, if the starting concentration of U in the FC is 10 mM, and 5 mM of these U has transported into the RC, then the permeation ratio here is 50%). Therefore, the permeation ratio (*R*) here is not equivalent to a *D*-value but it is similar to the so-called extraction efficiency/percentage (%) in solvent extraction. To calculate the *D*-value in this work, we have to convert the permeation ratio by using the equation $D = R/(1-R)$ (For example, a permeation ratio of 50% is equivalent to a *D*-value of 1, if with equal volumes in FC and RC). Then, the *SF* values are calculated by taking the ratio of *D* values of the two species we are trying to separate (*A_n* and *L_n* in this work).

Nevertheless, we believe the term "permeation ratio" in our previous manuscript might have misled the reviewer because this term looks like a "distribution ratio". To avoid such a

confusion, we have renamed the term “permeation ratio” as “permeation percentage (*Pct.*)” in our revised manuscript. Moreover, we have also defined the permeation percentage as well as the separation factor more clearly in our revised manuscript.

2) Page 7 Line 209-214: The discussion on the stable oxidation states of the actinides and their relationship to the permeability of the GOM is very weak. To begin with, in 3 M HNO₃ the preferred oxidation state of Np is a mixture of Np(V) and Np(VI), with the majority being Np(VI). Second, in the presence of NaBiO₃ both Np and Pu will be completely oxidized to the An(VI) state, along with Am (albeit, Am is a bit more difficult to oxidize). Finally, and most importantly, there was no evidence in the UV-Vis-NIR absorbance spectrum that anything but the An(VI) existed in the presence of NaBiO₃. In light of this, it seems a different reason is necessary to account for the An(VI) species crossing the GOM.

Response: We appreciate greatly the reviewer’s comment. First, we agree with the reviewer that the reduction of the An(VI) cannot account for the overall permeation of An across the GOM. Therefore, we attribute the permeation of An (5-10%) across the GOM under oxidizing condition mainly to the following reasons (see line 261-268 in the 1st revision version): *“We attribute this non-ideal rejection mainly to the intrinsic structural feature of the GOM. It is well known that pinhole defects exist on GO sheets and stacking voids are present in GOM due to the imperfect stacking of GO sheets, and these defects and voids are usually large in size and thus will enhance the liquid transport through the GOM and provide additional pathways for the permeation of larger ions such as actinyl in the GOM. Moreover, the linear configuration of actinyl ions is another factor need to be considered. The actinyl ions may enter the nanochannels of GOM in a horizontal position, which could ease the steric constraint and lead to permeation of these actinyl ions through the GOM.”* The discussion in line 209-214 was only to provide an explanation on the permeation trend (Am > Pu > Np ~ U, but the difference is within 3% of total permeation), which we believe is related to the redox properties of the four actinides. As pointed out by the reviewer, in 3 M HNO₃ the preferred oxidation state of Np is a mixture of Np(V) and Np(VI), which have also been proved through our additional experiments (**Figure R1** below). Both Np(V) and Np(VI) are actinyls and they can be well blocked by the GOM, thus resulting in a similar permeation behavior to U(VI). On the other hand, the final reduction products of Pu(VI) and Am(VI) are the spherical Pu(IV) and Am(III) ions, respectively, both of which will pass through the GOM along with lanthanides. It is true that the UV-Vis-NIR absorbance spectra have indicated that An(VI) can be quantitatively obtained and stabilized through NaBiO₃ oxidation, that’s also why we have stated in line 215-217 *“Nevertheless, the permeation ratios of the four actinides only differ slightly, implying that all the four actinides are well maintained in their actinyl form in the FC during the permeation process with the help of NaBiO₃ oxidation”*. However, it should be noted that there will still be slight reduction of An(VI), especially for Am(VI), over a long term contact with the GOM. As shown in **Figure R2**, around 2-3% of Am(VI) can be reduced to Am(III) in the presence of GO flakes and NaBiO₃ after 24 hours. Such an observation is well consistent with the slightly higher permeation percentage of Am than U in our permeation experiments.

Figure R1. Variation of Np absorption spectra in 3.0 M HNO₃ solution for 0-24 h. Initial Np concentration: ~0.2 mM Np(V).

Figure R2. The evolving of Am absorption spectra as a function of time in the presence of GO flakes and NaBiO₃. 1.5 mL of ~0.03 mM Am in 3.0 M HNO₃ solution, 22.5 mg NaBiO₃ and 1 mg GO flakes was added (This figure is also shown as Figure R17 in our response in the first-round revision).

Moreover, in our newly revised manuscript, we have introduced one more factor to account for the permeation of An(VI) through the GOM according to the comments of reviewer 1. As indicated by the broadened XRD peak, the distribution of the interlayer spacing between GO nanosheets in GOM is not uniform but should be scattered in a certain range. In this case, there must be interlayer spacing larger than the size of actinyl ions present in the GOM, providing additional channels for the permeation of An(VI). Moreover, to avoid misunderstanding in the discussion of the relationship between oxidation states of the actinides and their permeability through GOM, we have first emphasized that the main factors (pinhole defects and non-ideal stacking of GO nanosheets) to account for the permeation of actinides before the discussion.

3) Page 9 Line 269-279: The proposed mechanism given by the authors is that the ions diffuse through defect in the GOMs occurring where two GO nanosheets come together, which is depicted in Fig. 5. Yet, throughout the manuscript the size of the pores in the nanosheets is what is attributed to the separation. What is the actual root of the separation?

Response: We apologize for the confusion. As suggested by our discussion on permeation mechanism, we believe it is the unique interlayer spacing between GO nanosheets providing the separation selectivity, while the defects on the GO nanosheets and non-ideal stacking of GO in GOM create additional pathways for An(VI) permeation. We have revised Fig. 5 to reflect the permeation mechanism more accurately. Nevertheless, due to the complex physical and chemical interactions involved in the permeation process, we must acknowledge that we are not able to show every detail in this scheme of possible permeation routes (sectional view) for lanthanides/actinides in GOM.

Reviewer #3

Though the authors have given their response very explicitly with the support of few additional experiments, they could not bring out the novelty or any breakthrough finding in this article which can qualify its publication in Nature communication. Even after reading the revised version carefully, I have following concerns over its novelty claim.

1. As claimed in the abstract “Herein, we report a novel and efficient separation ………”, neither NaBiO_3 is a novel reagent for oxidation of Am(III) to Am(VI), nor the synthesis of GO with desired inter-layer spacing is new. Authors themselves have agreed that these materials are well studied. Then there is no overall novelty which fits for its publication in Nature Comm.

Response: We have made detailed replies on this issue in our previous response. Here we would like to further emphasize our viewpoints: we never claimed the oxidation with NaBiO_3 or the synthesis of GO with desired inter-layer spacing as novel in our manuscript, but it is the simple combination of oxidation, GO membrane sieving and solvent extraction giving a novel concept/method to separate An (U, Np, Pu, Am) from Ln in group under highly acidic conditions. We believe the novelty of science is not simply to create new materials or new reactions, it can also not be defined by “atypicality”. Novelty can be achieved by creating unexplored paths based on existing and classical knowledges. In this work, we achieved for the first time appreciable separation of An from Ln in group under highly acidic conditions through ion sieving in membranes, especially overcoming the seminal challenge of Am(VI) reduction in separation. Such an appreciable An/Ln group separation or even Am/Ln separation has never been achieved before by membrane techniques. We believe these findings in our work are of sufficient novelty and significance to afford its publication in Nat. Commun.

2. Authors claim for applicability of GO/TODGA assisted membrane for treating genuine nuclear waste (page 6 last para) is indeed a wrong statement. In fact, they are only targeting what is transported in the RC side. But, the actual product will remain in the FC side. This feed containing Am(VI) will always be contaminated with lanthanides as 100% transport of

lanthanides is not possible.

Response: Seemingly the reviewer has misinterpreted our words. In that context, we meant to “*further assess the applicability of our sieving technique for genuine waste treatment*”, so we explored the separation of a mixed actinides/lanthanides waste solution by sieving through the GOM. And the high separation factors in one stage sieving indicate our method is of potential applicability for real An/Ln waste treatment. We agree with the reviewer the actual product (An) will remain in the FC side and one stage of separation is not sufficient to give pure An product, so as we have stated clearly in our manuscript context as well as in our previous response to the reviewer’s comments, this strategy requires further improvements to enhance its practicability in real applications and we believe the separation efficiency can be further enhanced by optimizing the GOM or by employing multiple stage sieving (one stage of sieving filters out ~95% of lanthanide and two stages of sieving are expected to separate >99% of lanthanides). Our findings in this work point out an alternative direction to address the challenges in nuclear waste treatment and we expect with strong confidence that our work will inspire more experimental and theoretical efforts to deal with this problem.

3. Looking at 100 to 200 times higher concentration of lanthanides as compared to Am in the nuclear waste, even 1% left over lanthanides in the feed will give their concentration more than Am. So this approach will not be good for separation of lanthanides from actinides. Please note: Separation of Pu and U from lanthanides is not important as they can be achieved very easily. But the main concern is separation of Am(III) from lanthanides.

Response: First, we would like to point out that the total concentration of lanthanides is usually tens of times but not 100 – 200 times higher as compared to Am. For example, for the spent fuel unloaded from a low burn-up (30 MWd/kg·U) reactor, the total content of Ln is ~30 times higher than that of Am (*Elements*, 2006, 2, 343-349; *Solvent Extr. Ion Exch.*, 2009, 27, 97-106; *Prog. Part. Nucl. Phys.*, 2011, 66, 144-166). On the other hand, as we stated in our response to comment 2, Ln can be further separated by employing multiple stages of sieving. Two stages of sieving are expected to remove >99% of lanthanides from the actinides, leading to an Am product with little leftover of lanthanides.

4. Authors have not taken into account of Mo, Tc, Ru in the feed as the concentrations of these metal ions in nuclear waste are orders of magnitude higher than Am. Obviously, being oxyocations/oxyanions, these metal ions will not be transported, and will remain with the Am product in FC side. In such a scenario, the product Am will be having more amounts of these metals and non-transported lanthanides than Am itself. Under these conditions how one can claim “an efficient separation method”?

Response: Thanks for this comment but we would like to gently remind the reviewer that we are dealing with An/Ln separation in this work and the elements Tc, Mo, Ru (also Sr, Cs, Fe, Pd mentioned by this reviewer in the first round reviewing) are somewhat beyond the scope of our topic in this work. It is well known there are tens of elements in a spent fuel waste, we never attempted to separate the freshly dissolved spent fuel waste through our sieving method and we

sincerely acknowledge that we are not able to get rid of all the elements through this technique. Nevertheless, to ease the reviewer’s concern and acquire more knowledge on the permeation behaviors of Mo, Tc and Ru, here we did conduct an additional sieving experiment for these three elements (Note, Re was used to replace the radioactive Tc considering their similar properties). As shown in **Figure R3**, all the three elements show comparable permeation behavior to lanthanides (under conditions of no extraction) and they could permeate through the GOM. For Mo and Re, the permeation behavior can be well understood because the hydration diameters of MoO_4^{2-} and ReO_4^- are reported to be 7.7 Å and 7.04 Å (*J. Phys. Chem.*, 1959, 63, 1381-1387), respectively, which are well below the nanochannel size in the GOM in our work. For Ru, the interpretation of the permeation data is not that straightforward because the speciation of ruthenium in nitric acid solutions has been long known to be both complex and dynamic. Typically, ruthenium is present as a variety of Ru nitrosyl (RuNO^{3+}) nitrate and nitro complexes in nitric acid solutions. Anyhow, the permeation results for Ru shown in **Figure R3** under our experimental conditions indicated the size of hydrated Ru(III) species in nitric acid might be not large, which is possible because of the small ionic size of Ru ($r \sim 0.8$ Å, much smaller than that of Ln and An) and the dynamic conversion of different Ru species.

Figure R3. Permeation of Mo, Re and Ru through GOM. Initial concentration of each metal: 5 mM, $[\text{HNO}_3] = 3.0$ M. The starting form of the metals are $\text{Na}_2\text{MoO}_4 \cdot 4\text{H}_2\text{O}$ (solid), NH_4ReO_4 (solid), Ruthenium nitrosyl nitrate (in 3.0 M HNO_3) for Mo, Re, and Ru, respectively. No extraction was applied in the receiving side during the permeation process.

5. Since GO membrane acts as a membrane filter, membrane fouling has not been looked into. In fact, it will never work under nuclear waste condition and membrane pores will be choked in no time. This is an important aspect that must be looked into before proposing any membrane separation process. This aspect is certainly missing in this manuscript.

Response: We appreciate the reviewer’s comment and believe the reviewer did raise an important issue, but we would like to point out that this issue has not been missing in our manuscript and we have already looked into this issue in the first-round revision according to other reviewers’ comments. The results are presented in the section “Robustness of GOM” in

our manuscript, Figure S8-11 in the supporting information file, as well as in our responses to the comments of reviewer #1 and #4 in our previous response letter. Briefly, neither the presence of solid NaBiO_3 or the transport of An/Ln through the GOM will choke the membrane pores in our permeation experiments.

6. Authors might have tried this membrane for nano or ultra-nano filtration by applying pressure. In such a case, the permeate will be lanthanides and all other non-retained species, whereas product Am(VI) and other elements of similar size will be collected in the reject stream (as depicted in Fig. 1a). In this way, one may expect very fast separation and its practical utility.

Response: We appreciate the reviewer's comment. It is true the permeation rate could be enhanced by applying pressure in filtration. However, in this work we are not using a traditional filtration manner but adopting a natural diffusion manner for the separation, because the presence of solid NaBiO_3 will block the membrane in traditional filtrations. In this case of natural diffusion, it's not easy to apply additional pressure in the FC side. Alternatively, we applied a solvent extraction system in the RC side, which have greatly enhanced the permeation rate of lanthanides.

Considering above week points, I do not find this manuscript suitable for Nature Communication. However, as I mentioned in my previous recommendation, undoubtedly this is a nice piece of work, and can be published as a regular full article (not as a communication).

Response: We believe we have fully addressed the concerns of the reviewer. Nevertheless, to appreciate the points of debates with the reviewers, we opt in the Transparent Peer-Review that will share the reviewers' comments and our responses with the readers. Moreover, we would like to point out that the current version (as well as previous versions) of our work is indeed a regular full article but not a communication.

Reviewer #4

The authors have addressed all my concerns from the previous review in the responses to the reviewers document and revised manuscript. The authors have responded in a very organized and thoughtful manner. I suggest publishing the revised manuscript.

Response: We very much thank the reviewer for helping us improve our manuscript.

REVIEWERS' COMMENTS

Reviewer #2 (Remarks to the Author):

"Ion Sieving in Graphene Oxide Membrane Enables Efficient Actinides/Lanthanides Separation"
Zhipeng Wang, Liqin Huang, Xue Dong, Tong Wu, Qi Qing, Yuexiang Lu, Jing Chen, and Chao Xu,
Nat. Commun., NCOMMS-22-30550B

This manuscript has been revised a second time according to feedback given by four reviewers and has been improved. I see no reason why this article should not be published; I recommend this article for publication.

Response to reviewers' comments

Reviewer #2

This manuscript has been revised a second time according to feedback given by four reviewers and has been improved. I see no reason why this article should not be published; I recommend this article for publication.

Response: We highly appreciate the reviewer's comments.